# MicroRNAs Let-7b-5p and miR-24-3p as Potential Therapeutic Agents Targeting Pancreatic Cancer Stem Cells

**DOI:** 10.3390/ijms262211066

**Published:** 2025-11-15

**Authors:** Maricela Medrano-Silva, Eric Genaro Salmerón-Bárcenas, Elena Arechaga-Ocampo, Nicolas Villegas-Sepúlveda, Leopoldo Santos-Argumedo, Sonia Mayra Pérez-Tapia, Mayte Lizeth Padilla-Cristerna, Georgina Hernández-Montes, Gabriela Hernández-Galicia, Ana Beatriz Sánchez-Argáez, Paola Briseño-Díaz, Carmen Sánchez-Torres, Arturo Aguilar-Rojas, Andrea Martínez-Zayas, Miguel Vargas, Rosaura Hernández-Rivas

**Affiliations:** 1Departamento de Biomedicina Molecular, Centro de Investigación y de Estudios Avanzados del Instituto Politécnico Nacional (CINVESTAV-IPN), Apartado Postal 14-740, Ciudad de México 07360, Mexico; maricelamsil@gmail.com (M.M.-S.); eric.salmeron@cinvestav.mx (E.G.S.-B.); nvillega@cinvestav.mx (N.V.-S.); lesantos@cinvestav.mx (L.S.-A.); gabriela.hernandez@cinvestav.mx (G.H.-G.); ana.sanchez@cinvestav.mx (A.B.S.-A.); csanchez@cinvestav.mx (C.S.-T.); andrea.martinez@cinvestav.mx (A.M.-Z.); mavargas@cinvestav.mx (M.V.); 2Departamento de Ciencias Naturales, Universidad Autónoma Metropolitana Unidad Cuajimalpa, Ciudad de México 05348, Mexico; earechaga@cua.uam.mx; 3Unidad de Desarrollo e Investigación en Bioterapéuticos (UDIBI), Escuela Nacional de Ciencias Biológicas, Instituto Politécnico Nacional, Ciudad de México 11340, Mexico; mayra.perez@udibi.com.mx; 4Laboratorio Nacional de Vacunología y Virus Tropicales, Escuela Nacional de Ciencias Biológicas (ENCB), Instituto Politécnico Nacional, Ciudad de México 11340, Mexico; cristerna@gmail.com; 5Coordinación de la Investigación Científica, Red de Apoyo a la Investigación, Universidad Nacional Autónoma de México, Ciudad Universitaria, Ciudad de México 14080, Mexico; yinna@cic.unam.mx; 6Departamento de Genética y Biología Molecular, Centro de Investigación y de Estudios Avanzados del Instituto Politécnico Nacional (CINVESTAV-IPN), Apartado Postal 07360, Ciudad de México 07360, Mexico; paola.briseno@cinvestav.mx; 7Unidad de Investigación Médica en Medicina Reproductiva, Unidad Médica de Alta Especialidad en Ginecología y Obstetricia No. 4, Instituto Mexicano del Seguro Social (IMSS), Ciudad de México 01070, Mexico; arturolation@gmail.com

**Keywords:** pancreatic cancer stem cells, Let-7b-5p, miR-24-3p, transduction pathways, therapeutic, differentiation

## Abstract

Pancreatic cancer poses a major clinical challenge due to its aggressiveness, frequent recurrence, and limited response to current chemotherapeutic approaches. Cancer stem cells (CSCs), particularly pancreatic CSCs (PCSCs), are key drivers of tumor initiation, therapeutic resistance, and disease relapse. MicroRNAs (miRNAs) have emerged as critical regulators of CSC biology and influence self-renewal, pluripotency, and drug resistance through key signaling pathways. To identify PCSC-specific miRNAs, we enriched these cells using the pancreosphere culture method and isolated PCSC+ and PCSC− populations using FACS based on their expression of CD44, CD24, and CD133 surface markers. MicroRNA microarray analysis revealed 31 differentially expressed miRNAs (DEmiRNAs), of which 10 downregulated miRNAs were involved in pathways regulating pluripotency, including the Wnt/β-catenin, TGF-β, MAPK, and PI3K/AKT pathways. Then, 2 of these 10 DEmiRNAs, let-7b-5p and miR-24-3p, were selected for experimental validation. Their overexpression in PCSC+ cells inhibited these pathways, downregulated pluripotency factors, and induced differentiation into endocrine and exocrine phenotypes, as confirmed by RT-qPCR, Western blot, and RNA-seq. Functionally, each miRNA reduced sphere formation, increased gemcitabine sensitivity, and suppressed tumorigenicity in vivo, highlighting their potential as therapeutic candidates. Restoring tumor-suppressive miRNA expression may offer a novel strategy to overcome chemoresistance and improve outcomes in pancreatic cancer.

## 1. Introduction

Pancreatic ductal adenocarcinoma (PDAC) is the most common and most aggressive form of pancreatic cancer. Characterized by its infiltrative growth and resistance to conventional therapies, PDAC remains almost invariably deadly. The overall five-year survival rate is approximately 10% globally [1]. Unfortunately, early detection is particularly difficult, as symptoms are typically absent until the disease reaches an advanced or metastatic stage. As a result of this, most patients are diagnosed late in disease progression, limiting therapeutic options. Current treatments—surgery, chemotherapy, and radiotherapy—have limited efficacy, with frequent recurrence and poor long-term outcomes. Even among patients eligible for surgical resection, the majority of them relapse and die within five years of treatment [2]. Despite its limited efficacy, gemcitabine, a pyrimidine analog, has remained the standard chemotherapy for advanced PDAC for many years. However, only 23.8% of patients respond to treatment, largely due to the dense tumor stroma, poor drug penetration, and the development of chemoresistance [3]. Advances in understanding PDAC pathogenesis have led to new therapeutic strategies. Compared with gemcitabine alone, combination regimens such as FOLFIRINOX—a four-drug regimen that comprises folinic acid, 5-fluorouracil, irinotecan, and oxaliplatin—and nab-paclitaxel with gemcitabine significantly improved median survival from six to eleven months [3]. Despite these advances, overall survival for PDAC patients has improved marginally, underscoring the urgent need for novel therapeutic targets. Pancreatic cancer stem cells (PCSCs) contribute critically to tumor progression, metastasis, and resistance to chemotherapy and radiation. Although they represent less than 1% of the tumor mass, PCSCs can self-renew, differentiate, form tumors in vivo, and generate nonadherent tumor spheres in vitro [4,5]. Therefore, targeting PCSCs is essential for improving PDAC treatment outcomes. A deeper understanding and characterization of PCSCs are crucial for the development of effective therapies aimed at eradicating these resistant cells.

A network of transcription factors, epigenetic mechanisms, and signaling pathways regulates stem cell (SC) self-renewal and differentiation. Key pathways—including FGF2, LIF, TGF-β, BMP4, WNT, MAPK, and PI3K-AKT—control the expression of pluripotency transcription factors such as *OCT4*, *SOX2*, and *NANOG*. The coordinated interaction between these pathways and pluripotency factors is essential for maintaining SC identity and function [6,7]. For example, WNT signaling promotes the synthesis of *OCT4* and *NANOG* in pluripotent stem cells, including mouse embryonic stem cells (mESCs) and human ESCs (hESCs) [8]. BMP4 supports mESC maintenance by stimulating TGF-β signaling, inducing DNA-binding/differentiation (Id) proteins, and activating SMAD1/5/8. Conversely, Activin activates TGF-β signaling through *SMAD2* to maintain primed mESCs and hESCs [9]. Additionally, the PI3K/AKT pathway increases *NANOG* expression by inducing GSK3β expression, underscoring the critical role of these pathways in regulating pluripotency transcription factors and the maintenance, self-renewal, and differentiation of cancer stem cells [10].

Increasing evidence has shown that microRNAs (miRNAs) regulate the expression of mRNAs encoding effector proteins and receptor ligands involved in key signaling pathways that control cancer stem cell (CSC) maintenance and differentiation [11]. For instance, upregulation of pri-let-7a and pri-miR-200c expression inhibits Wnt/β-catenin signaling by targeting the 3′ UTRs of *PLAG1* and *HMGA2* [12]. Let-7 reduces the expression of the pluripotency factors *OCT4* and *SOX2*, and its levels increase in differentiated tumors, thereby suppressing stemness traits alongside those of the miR-200 family. Conversely, NF-κB signaling induces miR-205 expression to sustain pathway activation [13]. Given the pivotal role of miRNAs in regulating CSC properties, this study aimed to identify miRNAs enriched in PCSCs and to functionally characterize two candidates with potential therapeutic relevance.

## 2. Results

### 2.1. Identification, Sorting, and Characterization of PCSC-like Cells from the Pancreatic Cancer Cell Line PANC-1

We enriched PCSCs using an in vitro sphere formation assay with the PANC-1 cell line. Unlike non-stem cells, PCSCs survive and proliferate under nonadherent conditions, forming visible spheres by Day 3 and reaching a maximum size on Day 7 (Figure 1a). Compared with 2D culture, flow cytometry assays confirmed an increased proportion of PCSCs (CD24+CD44+CD133+triple-positive cells) in spheres [14]. As expected, triple-positive cells were enriched approximately 10-fold in spheres compared with those in 2D culture (Figure 1b). Using CD44, CD133, and CD24 markers, we identified and isolated PCSC+ (triple-positive) and PCSC− (double-negative: CD44+, CD133−, and CD24−) populations from pancreospheres using FACS (Figure 1c). Sorted PCSC+ and PCSC− cells were used in subsequent experiments. RT-qPCR confirmed that compared with PCSC− cells, PCSC+ cells overexpressed key pluripotency transcription factors—*SOX2*, *OCT4*, *NANOG*, *MYC*, and Klf4—which is consistent with their stemness profile (Figure 1d) [14].

CSCs are characterized by hyperactivation of key signaling pathways as well [15]. Western blot analysis revealed higher expression of effector proteins in the Wnt (β-catenin), MAPK (ERK1/2, RAS), PI3K/AKT (AKT, *MYC*), and TGF-β (*SMAD4*, *SMAD1*) pathways in the PCSC+ cells compared to PCSC− cells (Figure 1e).

Overall, flow cytometry, RT-qPCR, and Western blot assays confirm that the triple-positive cells are PCSC+ cells.

### 2.2. Identification of the miRNA Profile of PCSC-like Cells

To identify DEmiRNAs between PCSC+ and PCSC− cells, we performed Affymetrix miRNA 4.0 microarray analysis on RNA from two independent PCSC+ and two PCSC− samples. Using TAC software with a cutoff of *p* ≤ 0.0002, fold change ≥ 2 or ≤−2, and a false discovery rate (FDR) < 0.05, we identified 31 DEmiRNAs in PCSC+ cells compared to PCSC− cells (Table 1).

Notably, all 31 miRNAs were downregulated in the PCSC+ cells (Figure 2a, Table 1). For example, the expression of hsa-let-7b-5p was downregulated by 585.6-fold (Figure 2a). The expression profiles of the 31 DEmiRNAs identified in the PCSC+ cells allowed a clear distinction from PCSC− cells, as illustrated in the heatmap (Figure 2b). To validate the microarray data, 10 of the 31 DEmiRNAs (let-7b-5p, let-7a-5p, miR-24-3p, let-7e-5p, miR-103a-3p, miR-320a, miR-107, miR-15b-5p, and miR-210-3p) were randomly selected for RT-qPCR analysis. The results confirmed the microarray findings, which revealed consistently lower expression levels of these miRNAs in the PCSC+ cells compared to the PCSC– cells (Figure 2c).

To assess whether the 10 of 31 previously validated DEmiRNAs are also downregulated in primary cultures derived from PDAC patient samples, we enriched PCSC+ cells through sphere formation from two primary cultures—MGKRAS004 and MGKRAS005—followed by FACS-based isolation of triple-positive and double-negative cell populations. Compared with 2D-cultured cells, PCSC+ cells from MGKRAS004 were enriched 100-fold (Figure 2d), whereas those from MGKRAS005 were enriched 3.56-fold (Figure 2e). RNA was subsequently extracted from PCSC+ and PCSC− of these cells, and RT-qPCR analysis confirmed that triple-positive cells from primary cultures expressed higher levels of key pluripotency transcription factors (*OCT4, NANOG, AND SOX2*) compared with PCSC− cells (Figure 2f), supporting their identity as PCSCs.

The expression of 10 of the 31 previously validated DEmiRNAs was subsequently analyzed in both primary cultures. Nine of these miRNAs were consistently expressed at lower levels in PCSC+ cells compared to PCSC− cells in both the MGKRAS004 and the MGKRAS005 primary cultures (Figure 2g). These findings suggest that these miRNAs are also present in patient-derived samples and may represent promising therapeutic targets for the elimination of PCSCs in PDAC.

### 2.3. Of the 31 DEmiRNAs, 10 Are Involved in Signaling Pathways That Regulate Pluripotency and Differentiation in PCSCs

miRNAs typically destabilize their target mRNAs via posttranscriptional repression [16]. To elucidate the biological roles of these DEmiRNAs in PCSC+ cells, their target mRNAs were identified using miRTarBase Release 7.0 and TarBase v.8. A subset of validated targets is presented in Appendix A. To further support potential miRNA–mRNA interactions, in silico predictions were conducted using TargetScan and miRTargetLink 2.0. Representative examples of predicted binding sites between each DEmiRNA and its corresponding mRNA target within the 3′ UTR are shown in Appendix A.

Pathway enrichment analysis using the KEGG database revealed that the 31 DEmiRNAs are involved in multiple pathways, including cancer-related signaling, cell cycle regulation, and key transduction cascades such as WNT, MAPK, TGF-β, and mTOR (Figure 3a, left panel). In addition, these miRNAs were associated with signaling pathways regulating stem cell pluripotency (Figure 3a, left side) [17].

Focusing on this pathway, we identified ten DEmiRNAs—let-7b-5p, let-7a-5p, miR-23a-3p, miR-191-5p, let-7c-5p, miR-24-3p, let-7e-5p, miR-23b-3p, miR-103a-3p, and miR-1246—as key regulators within the pluripotency network (Table 2).

KEGG-based analysis of the predicted target mRNAs of these ten DEmiRNAs revealed multiple candidates encoding proteins involved in major signaling pathways, including MAPK, TGF-β, WNT, BMP, and PI3K-AKT (Figure 3a, right panel). These miRNAs appear to regulate a broad range of mRNAs encoding ligands, receptors, and downstream effectors within these pathways—particularly WNT, TGF-β, Nodal/Activin, BMP, MAPK, and PI3K-AKT—highlighting their potential role in modulating the signaling networks that govern stem cell pluripotency (Figure 3b, Table 3).

These pathways regulate CSC differentiation, pluripotency, and self-renewal [17]. Given that all ten miRNAs are downregulated and participate in pluripotency-maintaining pathways, we propose that their reduced expression allows the activation of genes essential for sustaining SC traits. To test this hypothesis, RT-qPCR was performed on six out of ten differentially expressed miRNAs implicated in pluripotency pathways (let-7a-5p, let-7b-5p, let-7c-5p, let-7e-5p, miR-24-3p, and miR-103a-3p). Our data confirm that these miRNAs are expressed at lower levels in PCSC+ cells and at higher levels in PCSC– cells, which is consistent with the results of the microarray (Figure 3c). Accordingly, if miRNA expression is reduced in PCSC+ cells, their target mRNAs are expected to be upregulated. RT-qPCR analysis supported this hypothesis, revealing elevated expression of key pathway-associated mRNAs—such as *FZD4*, *CDKN1*, and *CCND1* (WNT); *SMAD2* (Activin); *SMAD5* (BMP); *SMAD4* (Activin/BMP); *KRAS* (MAPK); and *AKT2* (PI3K/AKT))—in PCSC+ cells compared with that in PCSC– cells (Figure 3d). We also observed that the master pluripotency factors *OCT4*, *SOX2*, *NANOG KLF4* and *MYC* are coregulated with target mRNAs via these signaling pathways, contributing to the maintenance of PCSCs and inhibition of differentiation (Figure 3d). Collectively, our data suggest that the ten DEmiRNAs act as negative regulators of pluripotency. Thus, their increased expression may suppress the synthesis of stemness-related proteins, promote the differentiation of PCSC+ cells, and reduce the expression of core pluripotency factors.

### 2.4. Overexpression of Let-7b-5p and miR-24-3p Reduces the Expression of Key Pluripotency Factors and Their Target mRNAs, Promoting the Differentiation of PCSCs

To confirm the involvement of these miRNAs in PCSC+ cell differentiation, two of the ten DEmiRNAs associated with pluripotency pathways were selected: *let-7b-5p*, whose expression was most strongly downregulated (−585.62-fold), and *miR-24-3p*, whose role in PCSCs was previously unexplored at the time this study began. To evaluate their functional impact, synthetic mimics of let-7b-5p and miR-24-3p were individually transfected or cotransfected into PCSC+ cells. Forty-eight hours post-transfection, RNA was extracted, and RT-qPCR was performed to confirm miRNA overexpression and assess its effect on the mRNA levels of key pluripotency factors. Both miRNA mimics were successfully overexpressed (Figure 4a), and their individual overexpression led to reduced mRNA levels of key pluripotency factors (*OCT4*, *NANOG*, *SOX2*, *MYC*, and *KLF4*) (Figure 4b). However, cotransfection did not produce an additive effect.

We also assessed whether overexpression of each mimic—individually or combined—affects the expression of their predicted target mRNAs, as identified with TargetScan and miRTargetLink v2.0 (Appendix A). For instance, the 3′UTRs of *KRAS* and *HMGA2* are complementary to that of let-7b-5p, whereas *ABCC3* and *SMAD5* are predicted targets of miR-24-3p. Both miRNAs also share common targets, including *FZD4*, *SMAD2*, *AKT2*, *P21*, and *MYC* (Figure 4c). As expected, RT-qPCR confirmed that let-7b-5p overexpression reduced the expression of multiple target mRNAs involved in key signaling pathways: *FZD4*, *CTNNB1*, and *HMGA2* (Wnt/β-catenin); *SMAD2* (TGF-β); *ID1* (BMP); *KRAS*, *CCND1*, and *MYC* (MAPK); and *MYC* and *AKT2* (PI3K/AKT) (Figure 4d). Western blot assays confirmed the reduced expression of selected target proteins following miRNA overexpression (Figure 4e). Similar transcriptional and protein-level downregulation was observed for several miR-24-3p targets (Figure 4c,e,f). These findings support the hypothesis that overexpression of let-7b-5p or miR-24-3p downregulates key signaling pathways involved in maintaining PCSC pluripotency.

Finally, to assess whether miRNA overexpression promotes differentiation, RT-qPCR was used to measure the expression of differentiation markers in the pancreas (Figure 5a). Two days post transfection, the expression of genes associated with endocrine (*NGN3*), exocrine (*GATA6*), ductal (*SOX9*, *HNF6*), and acinar (*PTF1A*) lineages increased, indicating the initiation of pancreatic differentiation (Figure 5b). Western blot assays were performed to confirm these findings at the protein level. Total protein extracts were obtained from PCSC+ cells transfected with let-7b-5p, miR-24-3p, a non-related miRNA (NR-miRNA), or non-transfected PCSC+ controls (NT-PCSC+). Antibodies against *GATA4* (exocrine/acinar progenitors), *SOX9* (ductal/endocrine progenitors), and the pluripotency markers *SOX2* and *NANOG* were used. As shown in Figure 5c, compared with NT-PCSC+ and NR-miRNA controls, let-7b-5p and miR-24-3p overexpression increased *GATA4* and *SOX9* levels but reduced *SOX2* and *NANOG* expression. No synergistic effect was observed upon cotransfection; thus, only individually transfected cells were used in subsequent experiments. These results indicate that let-7b-5p and miR-24-3p independently promote PCSCs.

To assess the transcriptomic impact of miR-24-3p and let-7b-5p overexpression on PCSC+ differentiation, RNA-seq was performed on PCSC+ cells transfected with each mimic or non-transfected controls. Differentially expressed genes (DEGs) were identified using a *p*-value ≤ 0.05 and a fold change (FC) ≤ −2 or ≥2. In miR-24-3p-transfected cells, 315 DEGs were identified, of which 167 were upregulated and 148 downregulated. In let-7b-5p-transfected cells, 221 DEGs were found, with 148 upregulated and 73 downregulated (Appendix A, respectively). Gene Ontology (GO) analysis of the DEGs revealed enrichment in biological processes related to cell differentiation, development, and organogenesis for both miR-24-3p and let-7b-5p. Representative genes for each process are shown in Appendix A. Gene set enrichment analysis (GSEA) further demonstrated that the DEGs are predominantly associated with pathways involved in pancreas development, including both the endocrine and exocrine lineages (Appendix A). These transcriptomic findings are consistent with previous RT-qPCR and Western blot results (Figure 5b,c), supporting the conclusion that overexpression of miR-24-3p or let-7b-5p drives the differentiation of PCSCs toward the endocrine and exocrine pancreatic lineages.

### 2.5. Overexpression of Let-7b-5p and miR-24-3p In Vitro Reduces the PCSC Population and Pancreosphere Size and Leads to Decreased Cell Proliferation, Invasion, Migration, and Resistance to Gemcitabine

After confirming that let-7b-5p and miR-24-3p overexpression promotes PCSC+ differentiation, we assessed whether this also reduces the PCSC+ population. Flow cytometry was performed using antibodies against CD24, CD44, and CD133 to quantify the number of PCSC+ cells following transfection. As shown in Figure 6a, let-7b-5p overexpression reduced the PCSC+ population to 1.5%, compared with NT- and NR-mimic controls (4.12% and 3.98%, respectively), representing a ~72% decrease. Similarly, miR-24-3p overexpression reduced the percentage of PCSC+ cells to 1.93%, corresponding to a ~52% reduction. The results from two independent experiments are summarized in the graph (Figure 6a, bottom).

Our previous results indicated that compared with control NT-PCSC+ cells and cells transfected with NR-miRNA, cells overexpressing let-7b-5p or miR-24-3p form smaller spheres, likely due to a reduced PCSC+ population. To test this hypothesis, 5000 cells were seeded in ultralow adhesion plates, and the sphere diameters were measured on Days 1, 3, 5, and 7. As shown in Figure 6b, spheres began to form on Day 3 and reached their maximum diameter by Day 7. The control spheres averaged 75–100 µm in diameter, whereas those overexpressing let-7b-5p or miR-24-3p measured 50–75 µm. However, compared with NR-miRNA-overexpressing and NT-PCSC+ cells, cells overexpressing let-7b-5p produced smaller spheres. Additionally, compared with those in the controls, the number and size of colonies in the PCSC+ populations overexpressing each miRNA were significantly reduced by 50%, as shown in Figure 6c.

Given the decreased number of PCSC+ cells upon overexpression of miR-24-3p or let-7b-5p, we hypothesized that these cells would show reduced cell invasion and migration. To test this hypothesis, invasion and migration assays were performed on PCSC+ cells overexpressing each miRNA. Compared with that of NR-miRNA– and NTPCSC+ cells, invasion was significantly reduced by approximately 70% for miR-24-3p and 54% for let-7b-5p (Figure 6d). Compared with that of the controls, migration was also markedly decreased by 5.7-fold for miR-24-3p and 4.25-fold for let-7b-5p (Figure 6e).

Building on previous evidence that let-7b-5p and miR-24-3p enhance chemosensitivity in gastric cell lines [18,19], we evaluated their effects on chemoresistance in PCSC+ cells. Compared with non-transfected and non-related miRNA controls, overexpression of both miRNAs significantly increased sensitivity to gemcitabine. Specifically, miR-24-3p and let-7b-5p enhanced gemcitabine sensitivity by 5-fold and 3.5-fold, respectively (Figure 6f).

### 2.6. Pancreospheres Overexpressing miR-24-3p and Let-7b-5p Generated Smaller Tumors in Nu/Nu Mouse Xenograft Models

To assess the effects of let-7b-5p and miR-24-3p overexpression on tumor growth in vivo, PCSC+ cells were transduced separately with lentiviral particles encoding each miRNA (Appendix A). Their overexpression was confirmed using RT-qPCR (Appendix A). Among the six clones per miRNA, three miR-24-3p clones (miR-24-C1, miR-24-C8, and miR-24-F8N) and three let-7b-5p clones (let-7b-C2, let-7b-D10N, and let-7b-F8N) with the highest and most consistent expression were identified (Appendix A). Two clones per miRNA were then selected—C1 and C8 for miR-24-3p and C2 and D10N for let-7b-5p. RT-qPCR analysis revealed that overexpression significantly downregulated the expression of their respective master transcription factors, with the strongest effects on the expression of miR-24-C8 and let-7b-D10N (Appendix A). These findings establish two stable PCSC+ clones per miRNA with consistent overexpression.

A total of 1 × 10^4^ cells from the miR-24-C8 and let-7b-D10N clones were independently injected subcutaneously into Nu/Nu mice (Figure 7a).

As expected, the body weight of the mice remained unchanged compared with that of the mice injected with non-transfected PCSC+ cells (Figure 7b). After 10 weeks, the tumors derived from both miRNA-overexpressing clones were significantly smaller in size and volume (Figure 7c,d). Notably, tumor volume was reduced approximately threefold for miR-24-C8 and fivefold for let-7b-D10N relative to that of the controls, indicating a stronger tumor-suppressive effect of let-7b-5p (Figure 7e). These results highlight the therapeutic potential of these miRNAs against PCSC+ cells.

## 3. Discussion

The aggressiveness and poor prognosis of pancreatic cancer necessitate new therapies. Targeting CSCs, which drive tumor growth and treatment resistance, is promising. MicroRNAs regulate key CSC traits—self-renewal, drug resistance, pluripotency factors, and signaling pathways [20]. Understanding the role of miRNAs in pancreatic CSCs could reveal novel therapeutic targets.

To investigate PCSC regulation, we enriched, sorted, and characterized PCSCs from the PANC-1 cell line. Microarray analysis revealed 31 DEmiRNAs between PCSCs and non-PCSCs, with 10 key miRNAs (including let-7b-5p and miR-24-3p) involved in pluripotency pathways. KEGG analysis revealed that these miRNAs likely suppress multiple signaling cascades, such as the PI3K-AKT, MAPK, TGF-β (via Activin/Nodal and BMP), and Wnt cascades. Since pluripotent adult stem cells rely on pathways such as K-RAS, Hedgehog, TGF-β, mTOR, and Wnt to maintain self-renewal, tumor growth, invasion, and progression, and given that the identified miRNAs target ligands, receptors, and downstream effectors in these networks, our findings emphasize the critical role of miRNAs in modulating signaling pathways that sustain PCSC functions.

Furthermore, our data indicate that these pathways regulate master pluripotency transcription factors—*OCT4*, *SOX2*, and *NANOG*—which are critical for SC self-renewal and differentiation [11]. For example, in hESCs, Wnt signaling promotes *OCT4* and *NANOG* expression. Activin A, a member of the TGF-β family, increases the expression of *OCT4*, *NANOG*, NODAL, WNT3, bFGF, and FGF8 [21]. The TGF-β pathway, via BMP4, activates SMAD1/5/8 and represses ID proteins involved in differentiation. Additionally, the activation of PI3K/AKT and MEK/ERK induces the expression of genes that maintain pluripotency, including *OCT4, NANOG, SOX2*, and *MYC* [21,22,23].

Moreover, since these pathways are hyperactive in CSCs [24], our data suggest that the 10 identified DEmiRNAs are downregulated to maintain their activity. This promotes the expression of pluripotency and other key transcription factors, as confirmed by our RT-qPCR and Western blot analyses. These miRNAs likely act as tumor-suppressive negative regulators of pluripotency, as their upregulation targets ligands, receptors, and downstream effectors within these pathways. Consequently, their increased expression may inactivate these signaling cascades, reducing the expression of master pluripotency factors and related genes and leading to a decreased PCSC population, as observed in our study.

Among the ten DEmiRNAs identified in PCSCs, four—let-7a-5p, let-7b-5p, let-7c-5p, and let-7e-5p—belong to the let-7 family, which includes 10 mature miRNAs in humans. The let-7 family acts as a tumor suppressor across multiple cancers by inhibiting tumor growth and metastasis and promoting CSC differentiation [25]. For instance, the downregulation of let-7a-5p and let-7f-5p is correlated with enhanced tumor growth and metastasis due to the dysregulation of oncogenic pathways [25]. let-7a targets genes involved in epithelial-to-mesenchymal transition (EMT) and key signaling pathways, such as the Notch and IGF pathways, which are crucial for CSC maintenance and tumor progression [25]. Similarly, let-7g regulates oncogenes such as RAS and *MYC*, reducing cell survival and proliferation [26]. Notably, let-7 levels progressively decrease in PDAC patients as the disease progresses [27]. Taken together, the literature and our data suggest that these ten DEmiRNAs suppress effector proteins, ligands, and receptors, thereby inhibiting associated signaling pathways and promoting stem cell differentiation. The overexpression of let-7b-5p and miR-24-3p reduced the expression of pluripotency factors, induced differentiation, and decreased migration, invasion, and colony formation. This finding aligns with prior reports in liver cancer cells in which let-7b inhibited growth and invasion [28]. Our data also revealed that let-7b-5p suppresses Wnt signaling by downregulating the expression of the Wnt ligands and coreceptors FZD and DISHEVELED, as well as reducing *OCT4* and *NANOG* expression [25].

We also revealed that let-7b-5p inhibits c-MYC expression by targeting the PI3K-AKT and MAPK pathways, highlighting its key role in PCSC biology. These findings support the involvement of these pathways in PCSC metastasis, differentiation, and self-renewal [29]. Additionally, Wnt proteins and BMP inhibitors in the tumor microenvironment promote self-renewal in PCSCs [30]. Our data show that let-7b-5p-mediated inactivation of the Wnt and BMP4 pathways reduces the self-renewal capacity of PCSCs.

The PI3K/Akt pathway promotes CSC invasion and migration in prostate cancer [31]. Similarly, our results show that let-7b-5p overexpression silences AKT, significantly reducing PCSC invasion and migration. These findings underscore the pivotal role of PI3K/Akt in PDAC progression and PCSC dissemination.

Consistent with its tumor-suppressive role, miR-24-3p has been reported to inhibit cancer cell migration and invasion by targeting key regulators such as Prdx6 [32]. Our data revealed that it regulates key signaling pathways (Wnt, BMP, MAPK, and PI3K/AKT) in PCSCs, although the exact mechanisms controlling invasion and migration remain unclear. Overall, miR-24-3p functions as a tumor suppressor in PCSCs.

Cancer cells can develop chemoresistance by altering key signaling cascades, notably the EGFR/PI3K/PTEN/Akt/mTOR axis [33]. miRNAs are recognized as critical regulators of these pathways and act through the posttranscriptional control of gene expression [34]. *AKT2* has been shown to modulate chemoresistance by downregulating MDR1 and MRP1 in various malignancies, including ovarian, endometrial, breast, gliomas, and renal carcinomas [35,36,37,38,39]. In renal cell carcinoma, let-7b overexpression suppresses *AKT2*, enhances apoptosis, and increases sensitivity to 5-fluorouracil (5-FU) [40]. Similarly, in gastric cancer, let-7b mimics reverse multidrug resistance and restores chemosensitivity by downregulating c-MYC and Lin28 expression, thereby promoting CSC differentiation [41]. In line with these findings, our study demonstrated that ectopic let-7b expression in PCSC+ cells enhances gemcitabine sensitivity. Western blot analysis revealed a reduction in *AKT2* levels, suggesting that let-7b-5p sensitizes cells to PI3K/AKT pathway inhibition, which is consistent with previous observations in renal and gastric cancer models.

CSC-mediated chemoresistance is driven by multiple signaling pathways, including the PI3K/Akt, NF-κB, Wnt/β-catenin, and MAPK pathways [33]. The Wnt/β-catenin pathway regulates stemness-associated genes such as *CD44*, contributing to resistance. PI3K/Akt and NF-κB promote antiapoptotic signaling, whereas MAPK promotes survival and proliferation. Additionally, EMT, often regulated by miRNAs such as miR-24-3p, is linked to both chemoresistance and acquisition of a CSC phenotype.

Our findings support a model in which let-7b-5p and miR-24-3p cooperate to modulate gemcitabine resistance through coordinated regulation of these interconnected pathways. let-7b may act by targeting RAS, MEK, ERK, AKT, and C-MYC, while miR-24-3p may affect upstream receptors and effectors, including c-MYC, as confirmed by RT-qPCR and Western blot analyses. Although the dominant pathway remains to be elucidated, our data suggest that dual miRNA overexpression enhances gemcitabine sensitivity via suppression of multiple pathways.

Several miRNAs have been implicated in the regulation of CSC differentiation, often acting as tumor suppressors whose expression is increased during this process [42]. Let-7b has been shown to promote differentiation by targeting oncogenes and critical signaling pathways, including the Wnt/β-catenin pathway in the liver and colon [28,43]. Our findings expand on these findings by demonstrating that let-7b-5p not only modulates Wnt signaling but also downregulates components of the Activin/Nodal (TGF-β) pathway (*SMAD1*, *SMAD2*, *SMAD3*), the BMP pathway (*SMAD2*, *SMAD4*), the MAPK pathway (RAS, ERK), and the PI3K/AKT pathway (AKT), as confirmed by RT-qPCR and Western blot analyses. Collectively, these findings suggest that let-7b-5p inhibits multiple signaling cascades and their downstream transcription factors (e.g., *OCT4*, *SOX2*, *NANOG*, and *MYC*), thereby attenuating stemness and promoting differentiation.

Similarly, our bioinformatic predictions (miRTargetLink 2.0 and TargetScanHuman v8.0), which were supported by RT-qPCR and Western blot validation, indicated that miR-24-3p targets ligands, receptors, and effectors across the same four signaling pathways. This miRNA also downregulates key transcriptional regulators of pluripotency, contributing to CSC differentiation. Notably, owing to the significant overlap in target genes between let-7b-5p and miR-24-3p, we did not observe a synergistic effect on co-expression, as evidenced by our RT-qPCR data.

Collectively, our results support the hypothesis that let-7b-5p and miR-24-3p inhibit the expression of specific ligands, receptors, and downstream effectors in the Wnt, Nodal/Activin, BMP, MAPK, and PI3K/AKT signaling pathways, thereby promoting the differentiation of PCSCs. In the absence of these miRNAs, hyperactivation of these pathways sustains the expression of pluripotency-associated genes and maintains the self-renewal capacity of PCSCs. Conversely, elevated expression of let-7b-5p and miR-24-3p suppressed these pathways, leading to loss of stemness and induction of differentiation. Consequently, compared with control cells, PCSC+ cells exhibit increased sensitivity to gemcitabine, reduced frequency, diminished sphere-forming capacity, and reduced tumor formation in vivo (Figure 7f).

These results not only provide mechanistic insight into the miRNA-mediated regulation of PCSC plasticity but also suggest that restoring the expression of let-7b-5p and miR-24-3p may represent a promising therapeutic strategy to counteract stemness-driven drug resistance in pancreatic cancer.

## 4. Materials and Methods

### 4.1. Cell Culture of Pancreatic Cancer Cell Line PANC-1 and Primary Cell Lines MGKRAS004 and MGKRAS005

The human pancreatic cancer cell line PANC-1 (American Type Culture Collection, Manassas, VA, USA, ATCC^®^ CRL-1469™), derived from a metastatic adenocarcinoma of the pancreatic head, was used to generate PCSCs due to its ability to form tumors in athymic nude mice and pancreospheres [44]. Cells were cultured in high-glucose DMEM (Sigma, Burlington, NJ, USA, D5030) supplemented with 10% Fetal Bovine Serum (FBS; Gibco, Waltham, MA, USA, 10437028) and 1 × penicillin/streptomycin (Gibco, Waltham, MA, USA, 10378016), at 37 °C in a humidified 5% CO_2_ incubator. Primary pancreatic tumor cell lines MGKRAS004 and MGKRAS005 were cultured under similar conditions in DMEM with 10% FBS and 1% penicillin/streptomycin, as previously described by Briseño-Díaz P. [45].

### 4.2. Pancreospheres Formation

Ultra-low attachment surface 96-well plates (Sigma, Burlington, NJ, USA, CLS3474) were used to plate PDAC-1 cells (5000 cells/well) in serum-free media supplemented with FGF-2 (10 ng/mL) (Sigma, SRP4037) and EGF (20 ng/mL) (Sigma, Burlington, NJ, USA, E9644). The spheres were collected after seven days. Then to obtain individual cells, the spheres were briefly disintegrated with 1 mL TrypLE^TM^ Express enzyme (1_X_) (Gibco, Waltham, MA, USA, 12605028) at 37 °C for 20 min.

### 4.3. Fluorescence-Activated Cell Sorting (FACS)

A total of 2.5 × 10^5^ cells were blocked with 1 mL of PBS 1× supplemented with 10% FBS for 20 min at room temperature. After blocking, the supernatant was removed, and 100 µL of an antibody mixture—containing FITC anti-CD24 (Abcam, Cambridge, UK, ab30350), PE anti-CD44 (Abcam, Cambridge, UK, ab269300), APC anti-CD133 (Abcam, Cambridge, UK, ab253259), and Zombie NIR™ viability dye (BioLegend, San Diego, CA, USA, 423105)—diluted in PBS 1× with 1.5% FBS, was added. Cells were incubated for 20 min at room temperature in the dark. Following incubation, cells were washed twice with 1 mL of PBS 1× containing 2 mM EDTA and centrifuged at 1000 rpm for 5 min to pellet the cells. Before acquisition, samples were filtered through a 40-µm cell strainer to remove cell aggregates.

Controls included unstained cells (for autofluorescence), single-stained compensation beads (Invitrogen, Waltham, MA, USA, 01-3333-41), and Zombie NIR™-only stained cells (to distinguish live/dead populations). All samples were analyzed using a FACSAria™ Fusion Cell Sorter (BD Biosciences, San Jose, USA), and the PCSC+ and PCSC- cells were sorted.

### 4.4. RNA Extraction and cDNA Synthesis

Total RNA was extracted using TRIzol reagent (Invitrogen, Waltham, MA, USA, 15596026) following the manufacturer’s instructions [46]. Residual genomic DNA was removed by treatment with RNase-free DNase I (Thermo Fisher, Waltham, MA, USA, EN0521). For cDNA synthesis, 1.5 µg of DNase-treated RNA was reverse-transcribed using oligo (dT) primers (Thermo Fisher, Waltham, MA, USA, SO131), dNTPs (25 mM), and SuperScript II reverse transcriptase (Invitrogen, Waltham, MA, USA, 18064014). The reaction was incubated at 42 °C for 90 min, followed by enzyme inactivation at 70 °C for 5 min. cDNA was quantified and stored at −20 °C until use.

### 4.5. Quantitative PCR (qPCR)

qPCR was performed using Power SYBR^®^ Green PCR Master Mix 2× (Applied Biosystems, Waltham, MA, USA, 4367659). Each 10 µL reaction contained 5 µL of Master Mix, 2.6 µL of nuclease-free water, 0.4 µL of primer mix (10 µM), and 2 µL of cDNA (200 ng total). Reactions were run in duplicate from two independent biological replicates. Gene expression was quantified using the 2^−ΔΔCt^ method [47], with GAPDH as the internal control. Data were analyzed and visualized using GraphPad Prism v8.02 (GraphPad Software, San Diego, CA, USA) and are presented as mean ± SEM. Primer sequences and PCR conditions are listed in Table 4.

### 4.6. Total Protein Extraction and Western Blotting

Cells were lysed in Laemmli buffer (125 mM Tris-HCl pH 6.8, 2% SDS, 20% glycerol, 10% β-mercaptoethanol, 0.01% bromophenol blue) supplemented with protease (Complete™, Roche, Basel, Switzerland) and phosphatase (PhosSTOP™, Roche, Basel, Switzerland) inhibitors. Lysates were centrifuged at 14,000× *g* for 30 min at 4 °C. Equal amounts of protein (30 µg) were separated by SDS-PAGE and transferred to nitrocellulose membranes (GE Healthcare, Chicago, IL, USA, RPN303D).

Membranes were blocked for 2 h at room temperature in 5% non-fat dry milk in PBS-Tween 0.05%, followed by overnight incubation at 4 °C with primary antibodies (Table 5). After washing, membranes were incubated with secondary antibodies (Table 5) for 2 h at room temperature. Protein signals were visualized using SuperSignal™ West Femto substrate (Thermo Fisher, 34095) and imaged with the ChemiDoc™ MP System (Bio-Rad, Hercules, CA, USA).

### 4.7. miRNA Microarray Analysis

miRNA expression profiling was performed using the GeneChip™ miRNA 4.0 Array (Thermo Fisher Scientific, 902412), which includes probes for 2588 human miRNAs (miRBase v21) [48], following the protocol described by Félix et al. [49]. Data were processed using Transcriptome Analysis Console (TAC) software v4.0.2 (Affymetrix, Santa Clara, CA, USA), including quality control, normalization, and summarization. Differential expression analysis between triple-positive and triple-negative PCSC populations was conducted using the following thresholds: Fold Change (FC) > 2 or <−2, *p*-value < 0.0002, and FDR < 0.05.

### 4.8. Analysis of miRNA Differential Expression

Microarray data were normalized using the Robust Multi-array Average (RMA) method with background correction, as described by Irizarry et al. [50]. Differential expression analysis was performed using the limma package with an empirical Bayes approach and linear models [51]. Heatmaps were generated using R (https://cran.r-project.org, accessed on 7 August 2024), and DEmiRNAs were annotated based on the Affymetrix miRNA 4.0 annotation file (miRNA4_0-st v1.annotations.20160922.csv).

### 4.9. RT-qPCR Expression of miRNAs

miRNA from triple-positive and double-negative samples, were reverse-transcribed according to the protocol outlined by Cirera and modified by Alvarez-Hilario [52]. miRNA primers were designed using miRprimer V2.0 software (https://sourceforge.net/projects/mirprimer/ accessed on 8 July 2024), as described by Busk [53]. Relative miRNA expression was calculated using the 2^−∆∆Ct^ method [47] with normalization to RNU6 as housekeeping miRNA for triple-positive and double-negative samples, respectively.

### 4.10. miRNA Target Prediction and Pathway Analysis

Putative target genes of miR-24-3p and let-7b-5p were identified using miRTargetLink 2.0 and TargetScanHuman v8.0 [54,55]. Only genes supported by both predictive algorithms and experimental evidence were selected. Pathway enrichment analysis was performed using the KEGG database via the Enrichr platform [56], applying a significance threshold of *p* < 0.05.

### 4.11. miRNA Mimic Transfection

miRNA mimics for miR-24-3p (MC10737), let-7b-5p (MC11050), and a non-targeting negative control (AM17010) were obtained from Ambion (Thermo Fisher Scientific, Waltham, MA, USA). PANC-1 cells were seeded at 5 × 10^5^ cells/well in ultra-low attachment 6-well plates and incubated overnight. The next day, cells were transfected with 50 nM of each mimic using Lipofectamine™ 3000 (Invitrogen, Waltham, MA, USA, L3000008) according to the manufacturer’s protocol.

Briefly, 50 nM of mimic was diluted in 125 µL of Opti-MEM (Gibco, Waltham, MA, USA, 31985062), and separately, 5 µL of Lipofectamine 3000 was diluted in 125 µL of Opti-MEM. The solutions were mixed and incubated at room temperature for 20 min before being added to the cells. Plates were gently rocked to ensure even distribution, and cells were incubated at 37 °C with 5% CO_2_. After 48 h, cells were collected for downstream analyses.

### 4.12. Total RNA Extraction and RNA Sequencing

Total RNA was extracted using the RNeasy Mini Kit (QIAGEN, Hilden, Germany, 74104) and treated with DNase I (RNase-Free DNase Set, QIAGEN, 79254). RNA concentration was determined using a NanoDrop Lite spectrophotometer (Thermo Fisher Scientific, USA), and RNA integrity was assessed using Qubit and the Agilent 2100 Bioanalyzer (Agilent Technologies, USA). All samples had an RNA Integrity Number (RIN) > 8.

Two micrograms of total RNA per sample were used for mRNA sequencing, performed by Novogene (Sacramento, CA, USA). Poly-A selection was used for mRNA enrichment, and libraries were prepared and sequenced on the NovaSeq X Plus platform (Illumina, San Diego, CA, USA, PE150), generating ~12 Gb of raw data per sample.

### 4.13. RNA-Seq and Pathway Analysis

RNA-seq data analysis was performed using the Galaxy platform (v24.1.2) [57]. Raw data quality was evaluated with FastQC (v0.74+galaxy1). Reads were aligned to the human reference genome hg19 using HISAT2 (v2.2.1+galaxy1) with default settings [58]. Transcript assembly and quantification were conducted with FeatureCounts (v2.0.3+galaxy2) [59] using NCBI RefSeq annotations. DEGs were identified with DESeq2 (v2.11.40.8+galaxy) [60] using cutoffs of FC ≥ 2 and *p*-value < 0.05. Heatmaps were generated with the heatmap2 function from the R gplots package (v3.2.0+galaxy).

Functional enrichment and pathway analyses were performed using ShinyGO data base [61] (<0.05). GSEA (GSEA v4.3.3) was conducted using the GSEA software (Version 4.3.3) [62] with pancreatic differentiation-related gene sets, including GOBP_PANCREAS_DEVELOPMENT, GOBP_ENDOCRINE_PANCREAS_DEVELOPMENT, and HP_ABNORMALITY_OF_EXOCRINE_PANCREAS_PHYSIOLOGY, obtained from the Molecular Signatures Database.

### 4.14. Clonogenic Assays

Colony formation assays were performed as described by Ávila-López [63].

### 4.15. Migration and Invasion Assays

Cell migration was assessed using Transwell chambers (Costar, New York City, NY, USA, 3422). Briefly, 1 × 10^5^ triple-positive PANC-1 cells—non-transfected or transfected with miR-negative control, miR-24-3p, or let-7b-5p—were suspended in serum-free DMEM and seeded into the upper chamber. The lower chamber contained DMEM with 10% FBS as chemoattractant. After 24 h at 37 °C, migrated cells on the lower membrane surface were fixed with 4% PFA, stained with 0.5% crystal violet, and quantified by microscopy using ImageJ software V 1.54 k.

Invasion assays were performed similarly using Matrigel-coated Transwell inserts (Corning, New York City, NY, USA, CLS354230). A total of 2.5 × 10^5^ cells in serum-free media were seeded in the upper chamber; the lower chamber contained 10% FBS or no FBS as control. After 24 h, non-invading cells were removed, and invading cells were fixed with 4% PFA, stained with 0.1% Giemsa, and counted microscopically. All conditions were performed in triplicate.

### 4.16. MTT Assay

Cell viability was assessed using the MTT assay (Sigma, M5655) following the manufacturer’s instructions. Briefly, 1 × 10^3^ cells per well were seeded in 96-well plates and incubated for 24 h. Then, 10 µL of MTT reagent was added per well and incubated for 4 h at 37 °C. Afterward, 100 µL of detergent reagent (4 mM HCl, 0.1% NP-40 in isopropanol) was added, and plates were incubated overnight in the dark. Absorbance was measured at 570 nm to determine metabolic activity.

### 4.17. Lentiviral Transduction of PCSC+S

Lentiviral particles expressing human miRNAs hsa-let-7b-5p (UGAGGUAGUAGGUUGUGUGGUU, HLMIR0007) and hsa-miR-24-3p (UGGCUCAGUUCAGCAGGAACAG, HLMIR0410) were obtained from Sigma-Aldrich (Burlington, NJ, USA). PCSCs (5 × 10^4^ cells) were transduced with lentiviral particles in the presence of 8 µg/mL polybrene (Sigma, Burlington, NJ, USA, TR-1003) and incubated at 37 °C for 2–3 days. Stable clones were selected by adding puromycin (1 µg/mL; Sigma, Burlington, NJ, USA, P8833) to the culture medium, which was replaced every 3 days for 3 weeks. As a negative control, PCSCs were transduced with lentiviral particles encoding a non-targeting shRNA.

### 4.18. Xenograft Mouse Model

All animal experiments were approved by CINVESTAV’s Institutional Animal Care and Use Committee (IACUC) (protocol number: 0437-25). Six male NU/NU mice by group, aged 4–6 weeks, were used in the experiments. Six Super Mouse 750™ model boxes were used. Each Super Mouse 750™ box housed three mice, so two boxes were used for each group. A total of 1 × 10^4^ PCSC-PANC-1, PCSC-24-3p, and PCSC-let7b-5p cells were injected subcutaneously into the torsos of the mice. Three times a week following inoculation, the mice were checked to monitor the progress of the tumor and mouse weight. Once the tumors reached a volume of 1200 mm^3^, the mice were euthanized by CO_2_ inhalation to minimize animal suffering, and the tumors were removed. Tumor volume was calculated using the formula: Tumor volume = (long diameter × short diameter^2^)/2.

## 5. Conclusions

In summary, our findings identify let-7b-5p and miR-24-3p as promising therapeutic candidates capable of modulating key signaling pathways involved in PCSC maintenance and chemoresistance. By targeting multiple components of the Wnt, Nodal/Activin, BMP, MAPK, and PI3K/AKT pathways, these tumor-suppressive miRNAs promote PCSC differentiation, sensitize cells to gemcitabine, and reduce tumor volume. These results highlight the therapeutic potential of miRNA-based strategies to reprogram CSCs and improve treatment efficacy and clinical outcomes in PDAC.

## Figures and Tables

**Figure 1 ijms-26-11066-f001:**
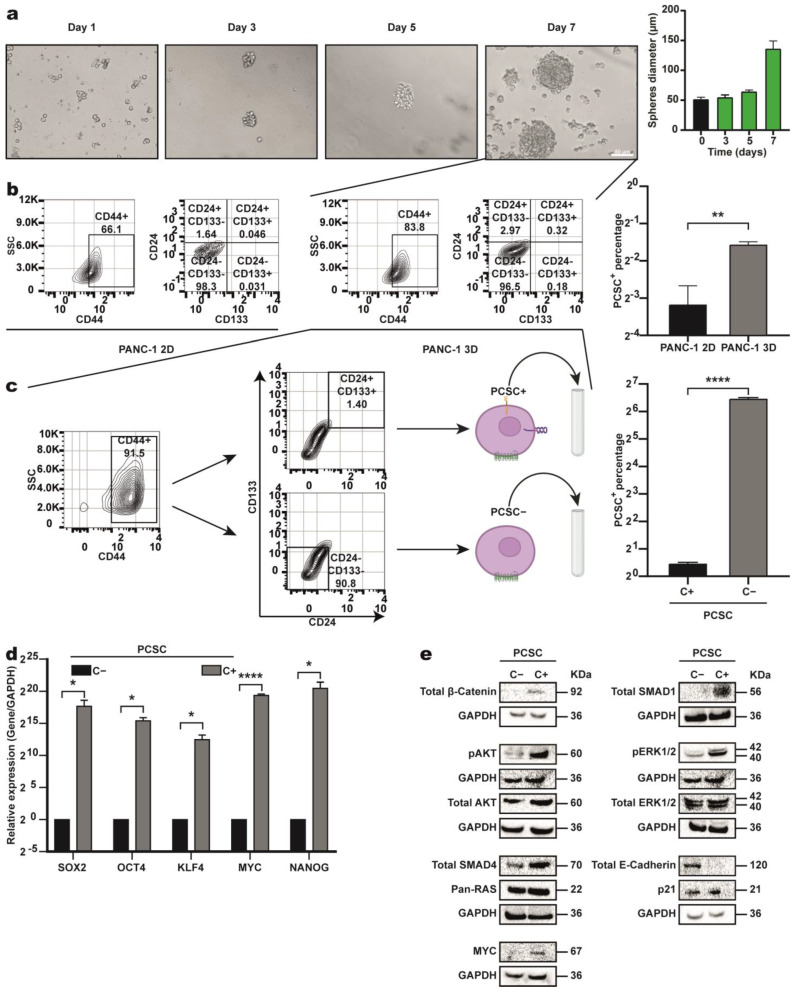
Identification, sorting, and characterization of PCSC-like cells from the PANC−1 cell line. (**a**) Growth of pancreospheres over 7 days; representative images at 50 µm. (**b**) Flow cytometry analysis of CD24+CD44+CD133+ cells in 2D vs. 3D cultures using specific fluorescent antibodies. Cells were stained with anti−CD24−FITC, anti-CD44−PE, anti-CD133−APC, and Zombie NIR™. (**c**) Sorting of PCSC+ (triple-positive) and PCSC− (double-negative) subpopulations from 3D cultures by FACS; representative plots and quantification. (**d**) RT−qPCR analysis shows increased expression of pluripotency markers (*SOX2, POU5F1, KLF4, MYC, NANOG)* in PCSC+ cells. Statistical significance was determined using two-tailed Student’s *t*-test (* *p* < 0.05; ** *p* < 0.01; **** *p* < 0.0001). The asterisks (*) indicate the level of statistical significance (**e**) Protein extracts analyzed by Western blot. PCSC+ corresponds to pancreatic cancer stem cells; PCSC− denotes non–pancreatic cancer stem cells; PANC-1 2D represents PANC−1 pancreatic cancer cells grown in a two-dimensional (2D) culture; and PANC1 3D refers to the same cell line grown in a three-dimensional (3D) culture.

**Figure 2 ijms-26-11066-f002:**
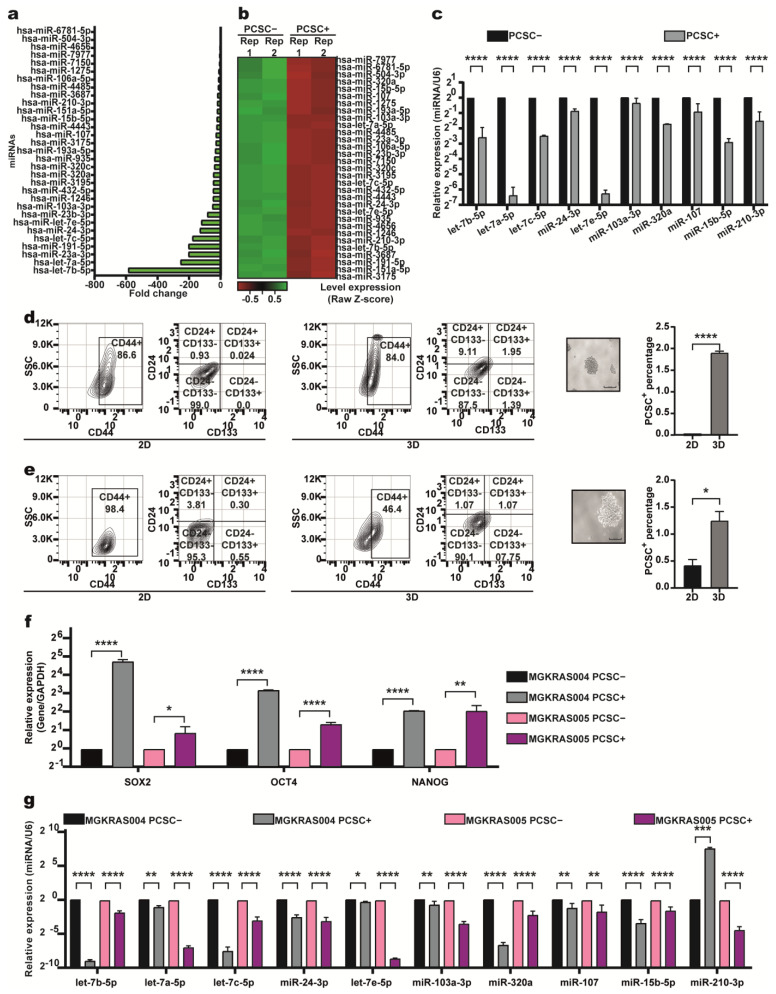
Thirty-one miRNAs are downregulated in PCSC+ cells from the PANC-1 cell line, and ten are also downregulated in PDAC patient-derived primary cultures. (**a**) Fold change analysis of 31 DEmiRNAs between PCSC+ and PCSC−cells. (**b**) Heatmap showing expression profiles of the 31 miRNAs in two PCSC+ and two PCSC−samples; red and green represent up- and downregulated miRNAs, respectively. (**c**) RT−qPCR validation confirms reduced expression of 10 selected miRNAs in PCSC+ cells. Statistical significance was assessed using two-tailed Student’s *t*-test (**** *p* < 0.0001). (**d**,**e**) Flow cytometry analysis of PCSC+ cells in MGKRAS004 and MGKRAS005 primary cultures under 2D and 3D conditions. Cells were stained with anti−CD24−FITC, anti-CD44−PE, anti-CD13−APC, and Zombie NIR™. Representative plots and quantification of PCSC+ cells are shown. (**f**) RT−qPCR analysis reveals higher expression of pluripotency transcription factors (*SOX2, OCT4, AND, NANOG*) in PCSC+ vs. PCSC− primary cultures. (**g**) Nine of the 31 DEmiRNAs are significantly downregulated in PCSC+ cells compared to PCSC− cells in both primary cultures. Statistical significance was determined using two-tailed Student’s *t*-test (* *p* < 0.05; ** *p* < 0.01; *** *p* < 0.001; **** *p* < 0.0001). The asterisks (*) indicate the level of statistical significance.

**Figure 3 ijms-26-11066-f003:**
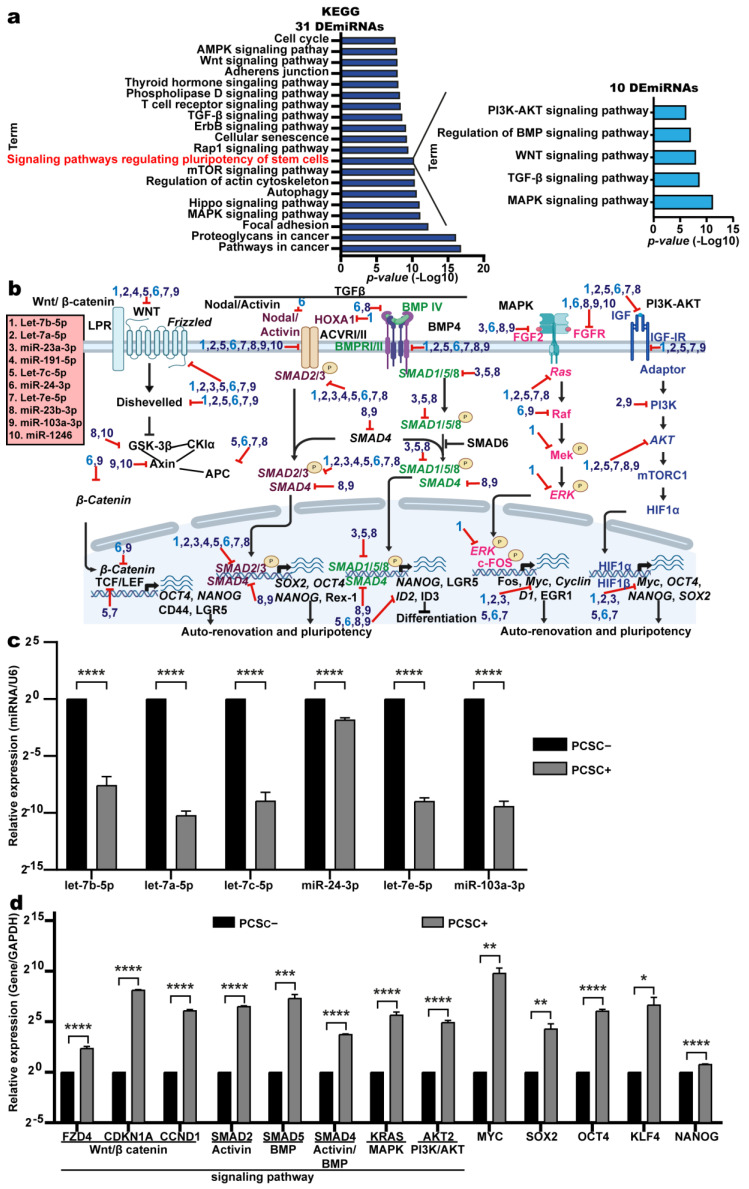
Modulation of stem cell pluripotency pathways by DEmiRNAs in PCSC+ cells. (**a**) KEGG pathway enrichment analysis of mRNA targets of the 31 DEmiRNAs identified in PCSC+ cells. These targets participate in cancer-related pathways, including WNT, MAPK, TGF-β, mTOR, and PI3K–AKT, as well as in signaling pathways regulating stem cell pluripotency. (**b**) Ten of the 31 DEmiRNAs modulate pancreatic cancer stem cell pluripotency by targeting key components of the WNT (black), Activin/Nodal (magenta), BMP (green), MAPK (pink), and PI3K–AKT (blue) pathways. Validated target genes (RT-qPCR and/or Western blot) are shown in italics. Let−7b−5p and miR−24−3p correspond to labels (1) and (**c**) RT-qPCR confirmed the downregulation of six DEmiRNAs in PCSC+ compared with PCSC− cells (**** *p* < 0.0001). (**d**) RT−qPCR analysis showed upregulation of selected mRNA targets in PCSC+ cells, including *FZD4*, *CDKN1A*, *CCND1*, *SMAD2*, *SMAD5*, *SMAD4*, *KRAS*, *AKT2*, *MYC*, *SOX2*, *POU5F1*, *KLF4*, and *NANOG*. Statistical significance: * *p* < 0.05; ** *p* < 0.01; *** *p* < 0.001; **** *p* < 0.0001 (two-tailed Student’s *t*-test).

**Figure 4 ijms-26-11066-f004:**
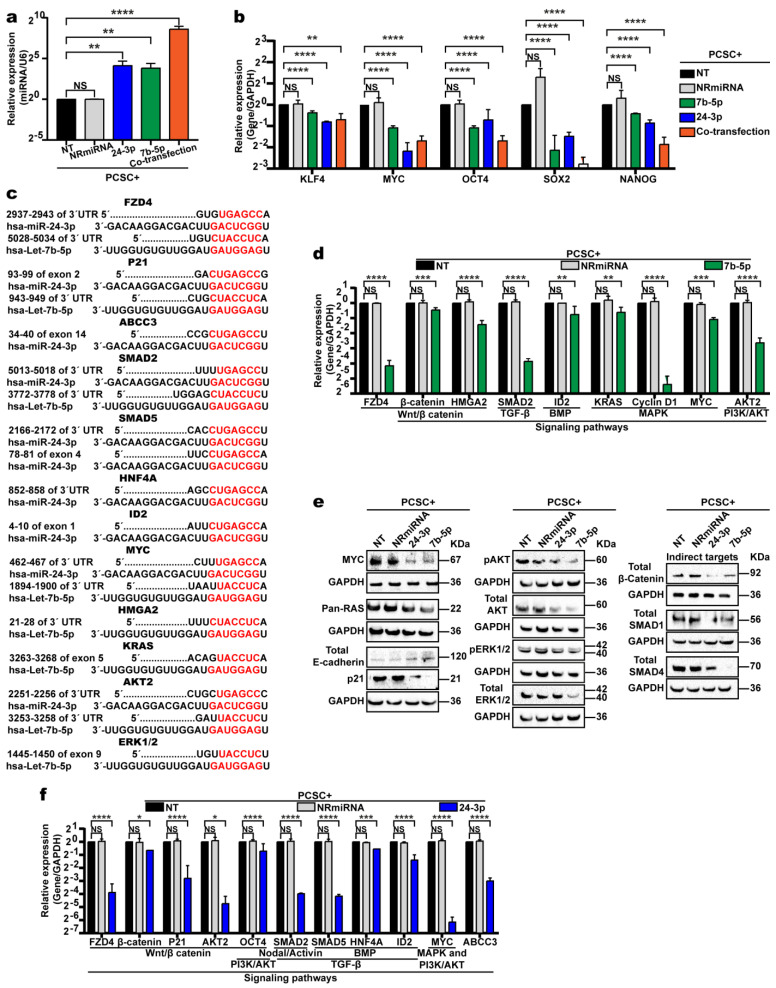
Let−7b−5p or miR−24−3p overexpression reduces signaling activity in PCSC+ cells. (**a**) Bar graph showing significant overexpression of let−7b-5p and/or miR−24−3p in PCSC+ cells transfected with the respective miRNAs, compared to non-transfected cells (PCSC-NT) or cells transfected with an unrelated miRNA (PCSC+-NRmiRNA). Two-tailed Student’s *t*-test. NS: not significant; ** *p* < 0.01; **** *p* < 0.0001. (**b**) RT−qPCR analysis reveals that miRNA overexpression, alone or combined, significantly reduces expression of *OCT4*, *MYC*, *KLF4*, *SOX2*, and *NANOG*. *p*-values as in (**a**). (**c**) Predicted mRNA targets were identified using TargetScan and miRTargetLink 2.0. (**d**) RT-qPCR shows that let−7b-5p overexpression decreases expression of targets involved in WNT/β-catenin, PI3K/AKT, TGF-β, BMP, and MAPK pathways (predicted by TargetScan, miRDB, and miRTargetLink 2.0), compared to PCSC-NT or PCSC− cells. ** *p* < 0.01; *** *p* < 0.001; **** *p* < 0.0001. (**e**) Western blot confirms reduced protein expression of shared target genes involved in these signaling pathways upon miRNA overexpression. (**f**) RT−qPCR assays showed that miR-24-3p overexpression similarly reduces the expression of predicted mRNA targets involved in the same pathways. * *p* < 0.05; *** *p* < 0.001; **** *p* < 0.0001. The asterisks (*) indicate the level of statistical significance.

**Figure 5 ijms-26-11066-f005:**
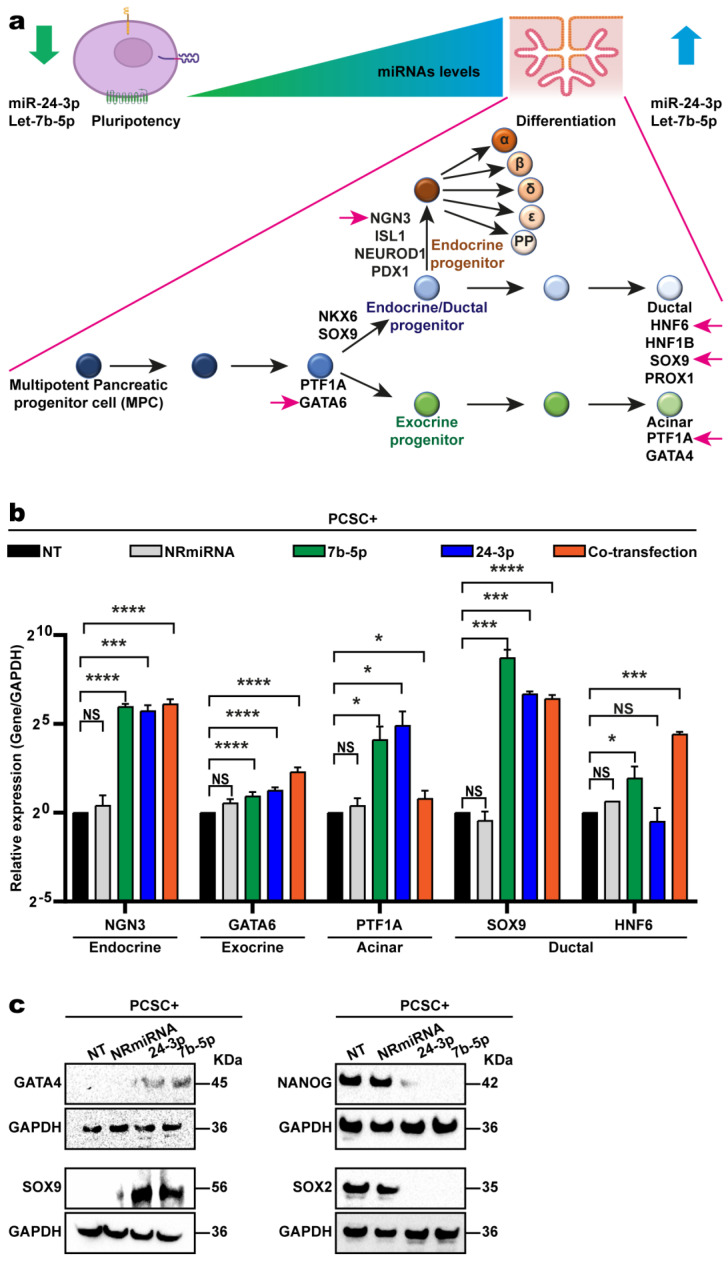
Let−7b−5p and miR−24−3p overexpression promotes stem cell differentiation in PCSCs. (**a**) Schematic representation showing that PCSCs, which express low levels of let−7b−5p and miR−24−3p, undergo differentiation upon overexpression of these miRNAs. This induces transcription of mRNAs associated with endocrine (*NGN3*), exocrine (*GATA6*), ductal-exocrine (*SOX9* and *HNF6*), and acinar-exocrine (*PTF1A*) progenitors. (**b**) RT-qPCR analysis of PCSCs transfected with let−7b−5p, miR−24−3p, or both, as well as non-transfected (NT-PCSC) and control (PCSC−) cells. Overexpression of one or both miRNAs upregulates mRNAs involved in endocrine and exocrine pancreatic differentiation. Two-tailed Student’s *t*-test. NS: not significant; * *p* < 0.05; *** *p* < 0.001; **** *p* < 0.0001. The asterisks (*) indicate the level of statistical significance. (**c**) Western blot analysis of protein extracts from transfected and control PCSC+s using antibodies against *GATA4* (acinar progenitors), *SOX9* (ductal progenitors), and pluripotency markers *SOX2* and *NANOG*.

**Figure 6 ijms-26-11066-f006:**
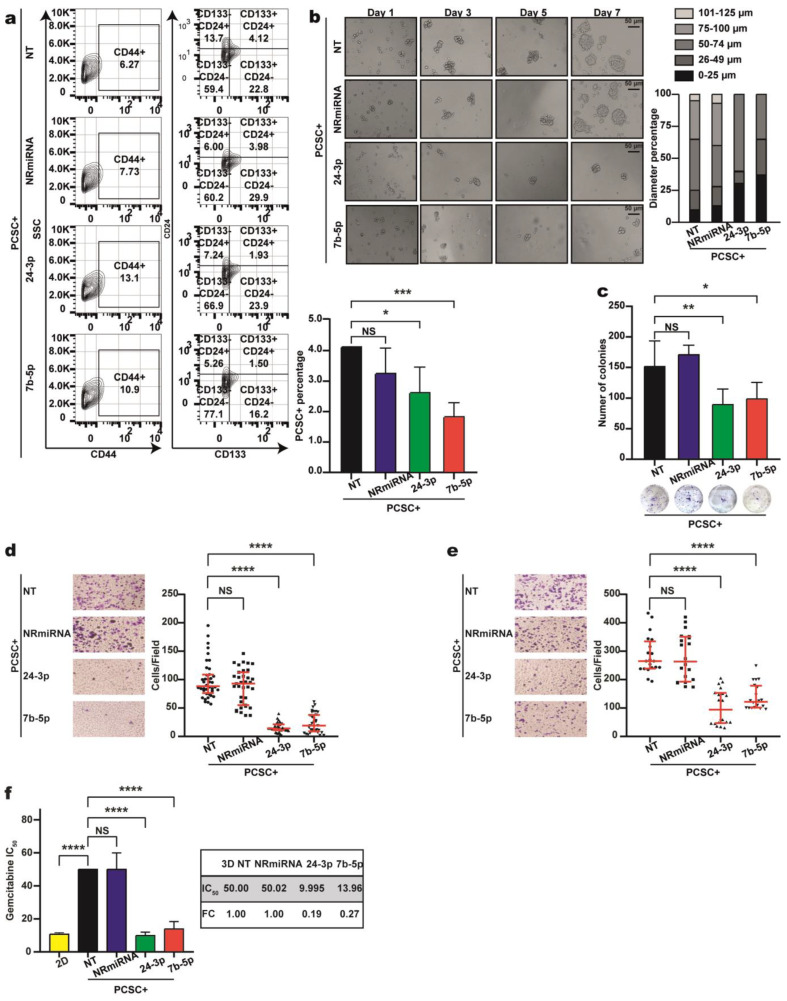
Let–7b–5p or miR-24–3p overexpression reduces the PCSC population, sphere formation, colony formation, migration, invasion, and gemcitabine resistance. (**a**) PCSCs were transfected with let–7b–5p, miR–24–3p, or a non-related miRNA (specificity control), or left non-transfected (NT-PCSC). Cells were labeled with anti-CD24–FITC, anti–CD44-PE, anti-CD133-APC, and Zombie NIR™. Flow cytometry identified CD24+CD44+CD133+ (triple-positive) cells. Representative plots show CD44 (**left**) and CD24/CD133 (right) expression. Significant reductions in the triple-positive population were observed in miRNA-transfected cells. Two-tailed Student’s *t*-test: NS: not significant; * *p* < 0.05; *** *p* < 0.001. (**b**) Overexpression of let–7b–5p or miR–24–3p reduced pancreosphere size over a 7-day culture period, compared to NT-PCSCs or cells transfected with non-related miRNA. Scale bar: 50 µm. (**c**) Clonogenic assays revealed fewer colonies in miRNA-overexpressing PCSCs than in control groups. Two-tailed Student’s *t*-test: NS; * *p* < 0.05; ** *p* < 0.01. (**d**,**e**) Migration and invasion assays showed that let-7b-5p or miR-24-3p transfection significantly reduced PCSC migratory and invasive capacity. Dot plots represent quantitative analysis. Two-tailed Student’s *t*-test: NS; **** *p* < 0.0001. (**f**) Cytotoxicity assays with gemcitabine (1–100 µM) showed increased sensitivity in miRNA-overexpressing PCSCs, as evidenced by lower IC_50_ values compared to controls. Two-tailed Student’s *t*-test: NS; **** *p* < 0.0001. The asterisks (*) indicate the level of statistical significance.

**Figure 7 ijms-26-11066-f007:**
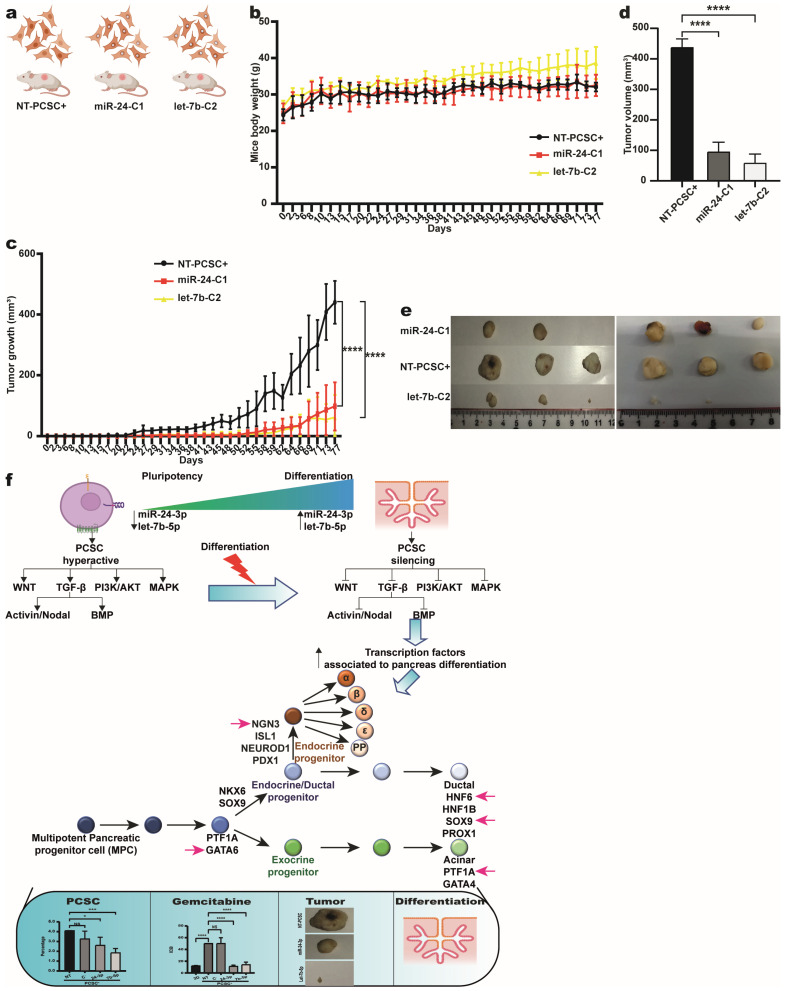
Let−7b−5p and miR−24−3p overexpression reduces tumor growth in vivo and regulates key signaling pathways in PCSCs. (**a**) Subcutaneous injections of 1 × 10^4^ cells from non-transduced PCSC-PANC-1 and miR-24−3p−C1 or let−7b−5p clones were administered independently to six mice per group. (**b**) Mouse body weight remained stable over 77 days with no significant differences among groups. (**c**) Tumor growth curves show significantly smaller tumors in mice injected with miR−24−3p−C8 or let−7b−5p−D10N clones compared to non-transduced PCSCs (One-way ANOVA), **** *p* < 0.0001). (**d**) Final tumor volumes at endpoint, presented as mean ± SEM (n = 6), are significantly reduced in miRNA-overexpressing groups (**** *p* < 0.0001). (**e**) Representative tumor images from mice injected with control PCSCs, miR−24−3p−C8, and let−7b−5p−D10N clones. (**f**) Let−7b−5p and miR−24−3p suppress key signaling pathways (Wnt, Nodal/Activin, BMP, MAPK, PI3K/AKT) by downregulating downstream effectors, promoting PCSC differentiation. Their absence leads to pathway hyperactivation, sustaining PCSC pluripotency and self-renewal. Overexpression inhibits these pathways, driving differentiation, reducing sphere and tumor formation, and increasing gemcitabine sensitivity.

**Table 1 ijms-26-11066-t001:** Identification of 31 DEmiRNAs in the PCSC.

miRNA	Fold Change	*p*-Value	FDR
hsa-let-7b-5p	−585.62	0.000014	0.006000
hsa-let-7a-5p	−253.85	0.000016	0.006000
hsa-miR-23a-3p	−203.30	0.000013	0.006000
hsa-miR-191-5p	−203.20	0.000007	0.006000
hsa-let-7c-5p	−178.18	0.000017	0.006400
hsa-miR-24-3p	−131.21	0.000023	0.006400
hsa-let-7e-5p	−121.77	0.000024	0.006500
hsa-miR-23b-3p	−83.93	0.000028	0.006800
hsa-miR-103a-3p	−51.81	0.000048	0.010500
hsa-miR-1246	−48.17	0.000048	0.010500
hsa-miR-432-5p	−47.51	0.000083	0.013400
hsa-miR-3195	−46.88	0.000088	0.013400
hsa-miR-320a	−46.15	0.000087	0.013400
hsa-miR-320c	−39.82	0.000069	0.013400
hsa-miR-935	−39.19	0.000093	0.013400
hsa-miR-193a-5p	−34.28	0.000098	0.013400
hsa-miR-3175	−30.31	0.000099	0.013600
hsa-miR-107	−29.59	0.000100	0.014000
hsa-miR-4443	−25.57	0.000100	0.014000
hsa-miR-15b-5p	−24.16	0.000100	0.014000
hsa-miR-151a-5p	−22.21	0.000100	0.014000
hsa-miR-210-3p	−21.12	0.000100	0.014000
hsa-miR-3687	−20.38	0.000100	0.014000
hsa-miR-4485	−15.77	0.000100	0.014000
hsa-miR-106a-5p	−14.78	0.000100	0.014000
hsa-miR-1275	−12.33	0.000100	0.014000
hsa-miR-7150	−8.25	0.000100	0.014000
hsa-miR-7977	−7.71	0.000100	0.014000
hsa-miR-4656	−5.63	0.000100	0.014000
hsa-miR-504-3p	−4.32	0.000200	0.014500
hsa-miR-6781-5p	−3.21	0.000200	0.014600

**Table 2 ijms-26-11066-t002:** 10 DEmiRNAs implicated in regulating stem cell pluripotency.

miRNA	Fold Change	*p*-Value	FDR
hsa-let-7b-5p	−585.62	0.000014	0.006000
hsa-let-7a-5p	−253.85	0.000016	0.006000
hsa-miR-23a-3p	−203.30	0.000013	0.006000
hsa-miR-191-5p	−203.20	0.000007	0.006000
hsa-let-7c-5p	−178.18	0.000017	0.006400
hsa-miR-24-3p	−131.21	0.000023	0.006400
hsa-let-7e-5p	−121.77	0.000024	0.006500
hsa-miR-23b-3p	−83.93	0.000028	0.006800
hsa-miR-103a-3p	−51.81	0.000048	0.010500
hsa-miR-1246	−48.17	0.000048	0.010500

**Table 3 ijms-26-11066-t003:** Potential mRNAs targets of 10 DEmiRNAs.

Let-7b-5p	Let−7a-5p	miR−23a−3p	miR-−191−5p	Let−7c−5p	miR−24−3p	Let−7e−5p	miR−23b−3p	miR−103a−3p	miR−1246
ACVR1B	ACVR1B	FZD5	BMI1	ACVR1B	ACVR1B	ACVR1B	ACVR1B	ACVR1	ACVR1
ACVR1C	ACVR1C	MYC	FZD5	ACVR1C	ACVR2B	ACVR1C	ACVR1C	ACVR1B	AXIN2
ACVR2A	ACVR2A	PIK3R1	INHBA	ACVR2A	APC	ACVR2A	ACVR2B	ACVR1C	DLX5
ACVR2B	ACVR2B	SMAD3	LIFR	ACVR2B	BMP4	ACVR2B	BMP4	ACVR2A	FGFR2
AKT2	AKT2	SMAD5	PCGF3	AKT2	BMPR1B	AKT2	BMPR1A	ACVR2B	FZD3
AKT3	AKT3	STAT3	PIK3R1	AKT3	BMPR2	AKT3	BMPR2	AKT2	GSK3B
APC2	APC2		SMAD2	APC2	DVL3	APC2	FGF2	AKT3	JARID2
BMPR1A	BMPR1A		WNT10B	BMPR1A	ESRRB	BMPR1A	FGFR1	APC	PIK3CA
DVL3	BMPR1B		WNT5A	DUSP9	FGF2	DUSP9	FGFR2	APC2	SKIL
FZD3	DLX5			DVL3	FGFR3	DVL3	FZD3	AXIN2	
FZD4	DUSP9			FZD3	FZD4	FZD3	FZD4	BMI1	
FZD9	DVL3			FZD4	FZD5	FZD4	FZD5	BMPR1B	
HAND1	FZD3			FZD8	GSK3B	FZD9	FZD7	BMPR2	
HOXA1	FZD4			FZD9	ID4	HAND1	GRB2	CTNNB1	
HOXB1	FZD9			HAND1	IGF1	HOXA1	GSK3B	DLX5	
HOXD1	HAND1			HOXA1	JAK1	HOXB1	HESX1	DVL1	
HRAS	HOXA1			HOXB1	JARID2	HOXD1	HOXA1	ESRRB	
IGF1	HOXB1			HOXD1	KAT6A	IGF1	ID4	FGF2	
IGF1R	HOXD1			ID1	MAPK14	IGF1R	IGF1	FGFR1	
KRAS	HRAS			IGF1	MYC	KAT6A	INHBA	FGFR2	
LEFTY1	IGF1			IGF1R	NODAL	KRAS	INHBE	FGFR3	
MYC	IGF1R			JARID2	ONECUT1	LEFTY1	ISL1	FZD10	
NRAS	KRAS			KRAS	PAX6	MYC	JAK1	FZD4	
ONECUT1	LEFTY1			LEFTY1	PCGF5	NRAS	JARID2	FZD6	
PCGF3	MAPK11			MYC	PCGF6	ONECUT1	KRAS	FZD7	
PCGF5	MEIS1			NRAS	PIK3R3	PCGF3	MAPK14	GSK3B	
PIK3CA	MYC			ONECUT1	RAF1	PCGF5	MEIS1	ID2	
SKIL	NRAS			PCGF3	SMAD2	PIK3CA	OTX1	IGF1R	
SMAD2	ONECUT1			PCGF5	SMAD5	SKIL	PCGF1	INHBA	
SMARCAD1	PCGF3			PIK3CA	SMAD9	SMAD2	PIK3CA	INHBB	
STAT3	PCGF5			SETDB1	TCF7	SMARCAD1	PIK3CB	JAK1	
WNT1	PIK3CA			SKIL	WNT10A	STAT3	PIK3R1	JARID2	
WNT11	PIK3R1			SMAD2	WNT11	WNT1	PIK3R3	KLF4	
WNT9A	SKIL			SMARCAD1	WNT2B	WNT9A	RAF1	OTX1	
WNT9B	SMAD2			STAT3	WNT4	WNT9B	REST	PAX6	
	SMARCAD1			WNT1	WNT8B		SKIL	PCGF2	
	STAT3			WNT9A	ZFHX3		SMAD2	PCGF3	
	WNT1			WNT9B			SMAD3	PCGF5	
	WNT9A						SMAD4	PIK3CD	
	WNT9B						SMAD5	PIK3R1	
							SMARCAD1	PIK3R3	
							ZFHX3	RAF1	
							ZIC3	REST	
								SMAD4	
								SMAD5	
								TCF3	
								WNT16	
								WNT3A	
								WNT4	
								WNT5A	
								WNT6	
								WNT8B	
								ZFHX3	

**Table 4 ijms-26-11066-t004:** Sequences of primers used in this study.

Target	Forward (5’ to 3’)	Reverse (5’ to 3’)	Ta (°C)
SOX2	GGATAAGTACACGCTGCCCG	ATGTGCGCGTAACTGTCCAT	55
OCT4	CACTGCAGCAGATCAGCCA	TGTGCATAGTCGCTGCTTGA	55
NANOG	TGTCTTCTGCTGAGATGCCTCACA	CCTTCTGCGTCACACCATTGCTAT	63
KLF4	GCAATATAAGCATAAAAGATCACCT	AACCAAGACTCACCAAGCACC	59
MYC	GAACTATGACCTCGACTACGACTC	GCAGATGAAACTCTGGTTCACCATG	55
NGN3	GGTAGAAAGGATGACGCCTC	CCGAGTTGAGGTCGTGCAT	51
GATA6	TCCCCCACAACACAACCTAC	GTAGAGCCCATCTTGACCCG	60
PTF1A	GAAGGTCATCATCTGCCATCGG	CCTTGAGTTGTTTTTCATCAGTCC	60
SOX9	AGGAAGCTCGCGGACCAGTAC	GGTGGTCCTTCTTGTGCTGCAC	63
HNF6	GAGGATGTGGAAGTGGCTGCAG	CTGTGAAGACCAACCTGGGCTT	55
CCND1	ACCATCCAGTGACAAACCAT	GTAGCGTATCGTAGGAGTGG	53
KRAS	AAGGCCTGCTGAAAAATGAC	TGGTCCTGCACCAGTAATATG	55
HMGA2	GAAGCCACTGGAGAAAAACGGC	GGCAGACTCTTGTGAGGATGTC	60
HNF4A	GGCTGCAAGGGCTTCTTC	GGCACTGGTTCCTCTTGTCT	51
CDKN1A	ACTCTCAGGGTCGAAAACGG	AGATGTAGAGCGGGCCTTTG	53
ABCC3	GAGGAGAAAGCAGCCATTGGCA	TCCAATGGCAGCCGCACTTTGA	60
SMAD5	CAGGAGTTTGCTCAGCTTCTGG	GGTGCTGGTTACATCCTGCCG	60
SMAD4	CTACCAGCACTGCCAACTTTCC	CCTGATGCTATCTGCAACAGTCC	63
SMAD2	GGGTTTTGAAGCCGTCTATCAGC	CCAACCACTGTAGAGGTCCATTC	65
FZD4	TTCACACCGCTCATCCAGTACG	ACGGGTTCACAGCGTCTCTTGA	63
AKT2	CATCCTCATGGAAGAGATCCGC	GAGGAAGAACCTGTGCTCCATG	63
ID2	TTGTCAGCCTGCATCACCAGAG	AGCCACACAGTGCTTTGCTGTC	63
CTNNB1	CACAAGCAGAGTGCTGAAGGTG	GATTCCTGAGAGTCCAAAGACAG	60
GAPDH	TTGTCAAGCTCATTTCCTGG	TGATGGTACATGACAAGGTG	60
miR-Let-7b-5p	CAGTGAGGTAGTAGGTTGTGT	GGTCCAGTTTTTTTTTTTTTTTAACCA	55
miR-Let-7a-5p	GCAGTGAGGTAGTAGGTTG	GGTCCAGTTTTTTTTTTTTTTTAACTATAC	53
miR-Let-7c-5p	GCAGTGAGGTAGTAGGTTGT	GGTCCAGTTTTTTTTTTTTTTTAACCA	55
miR-24-3p	AGTGGCTCAGTTCAGCA	GTCCAGTTTTTTTTTTTTTTTCTGTTC	47
miR-Let-7e-5p	GCAGTGAGGTAGGAGGTTG	GGTCCAGTTTTTTTTTTTTTTTAACTATAC	55
miR-103a-3p	GCAGAGCAGCATTGTACAG	GGTCCAGTTTTTTTTTTTTTTTCATAG	60
miR-320a	CAGAAAAGCTGGGTTGAGA	CAGTTTTTTTTTTTTTTTCGCCCT	55
miR-107	GCAGAGCAGCATTGTACAG	GGTCCAGTTTTTTTTTTTTTTTGATAG	60
miR-15b-5p	GCAGTAGCAGCACATCA	CCAGTTTTTTTTTTTTTTTGTAAACCA	47
miR-210-3p	GCGCAGCTGTGCGTGTGACA	GTTTTTTTTTTTTTTTCAGCCGCT	60
RNU6	CTCGCTTCGGCAGCACATATACT	ACGCTTCACGAATTTGCGTGTC	60
RT-primer	CAGGTCCAGTTTTTTTTTTTTTTTVN	-	-

**Table 5 ijms-26-11066-t005:** Antibodies used in this study.

Antibody	Molecular Weight (kDa)	Source	Dilution	Assay	Catalogue Number	Manufacturer
β-Catenin	92	Rabbit	1:1000	WB	#9582	Cell Signaling
SMAD1	52–56	Mouse	1:200	WB	SC-7965	Santa Cruz
pAKT	60	Rabbit	1:2000	WB	#4060	Cell Signaling
AKT	60	Rabbit	1:1000	WB	#9272	Cell Signaling
pERK1/2	42, 44	Rabbit	1:1000	WB	#9101	Cell Signaling
ERK1/2	42, 44	Rabbit	1:1000	WB	#9102	Cell Signaling
SMAD4	70	Rabbit	1:1000	WB	#46535	Cell Signaling
Pan-RAS	21	Mouse	1:250	WB	AESA02	Cytoskeleton
E-cadherin	80, 120	Mouse	1:750	WB	SC-8426	Santa Cruz
p21	21	Mouse	1:500	WB	GTX629543	GeneTex
Myc	67	Mouse	1:1000	WB	626802	BioLegends
Sox9	56	Rabbit	1:1000	WB	185966	Abcam
GATA4	45	Mouse	1:100	WB	SC-25310	Santa Cruz
Nanog	42	Rabbit	1:2000	WB	#4903	Cell Signaling
SOX2	35	Rabbit	1:1000	WB	#3579	Cell Signaling
GAPDH	36	Rabbit	1:80,000	WB	GTX100118	GeneTex
Mouse-HRP	-	Rabbit	1:5000	WB	SAB3701023	Sigma
Rabbit-HRP	-	goat	1:5000	WB	A0545	Sigma
CD24-FITC	35–45	Mouse	1:100	FC	555427	BD Pharmingen
CD44-PE	80–95	Mouse	1:100	FC	555479	BD Pharmingen
CD133/1-APC	120	Mouse	1:100	FC	130-113-668	Miltenyi Biotec
CD24-FITC	30–50	Mouse	1:100	FC	ab30350	Abcam
CD44-PE	81	Mouse	1:100	FC	ab269300	Abcam
CD133-APC	97	Mouse	1:100	FC	ab253259	Abcam

WB: Western blot; FC: Flow Cytometry.

## Data Availability

The data supporting the conclusions of this article are available in NCBI’s Gene Expression Omnibus (GEO) and are accessible through GEO accession numbers GSE294243 and GSE294244.

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
