# Peer review of "MicroRNAs Let-7b-5p and miR-24-3p as Potential Therapeutic Agents Targeting Pancreatic Cancer Stem Cells"

_ijms, 2025, doi:10.3390/ijms262211066_

Round 1
Reviewer 1 Report
Comments and Suggestions for Authors
Researchers have identified the MicroRNAs let-7b-5p and miR-24-3p as potential therapeutic agents targeting pancreatic cancer stem cells. These findings are supported by figures illustrating expression patterns, functional assays, and proposed mechanisms.
Fig. 1c. The drawing appears to have been created by hand rather than using software. Please clarify this. Additionally, include abbreviations in a separate legend.
Fig. 2a. The labels are unclear. Similarly, Figs. 2d and 2e look like they were drawn by hand rather than generated by software. Please clarify these figures as well.
Fig. 3. The labels in figures 3a-g remain unclear. Please redraw Fig. 3b with simplified flow arrows and emphasize the novel findings.
Fig. 4. The labels in the figures are still unclear.
Fig. 6a. This figure has the same issue as Fig. 1c.
In summary, the manuscript presents a comprehensive combination of biochemical and cellular approaches. There is a logical flow from molecular mechanisms to biological relevance, and the figures collectively support the central hypothesis.
The manuscript has solid experimental depth and could be suitable for IJMS after major figure and presentation revisions—specifically improving data quantification, visual clarity, and the conceptual model.
need to be improved.
Reviewer 2 Report
Comments and Suggestions for Authors
I did some corrections on the original manuscript as in the uploaded file
Minor comments
Line 86 Space [6,7]
Line 89 (id) is an incorrect abbreviation, and Smad 1/5/8 is correct, where in all text mentioned smad 2/4/5
Line 124 space as in the edited text
Line 126 makes the tumorigenecity assay first, to be with the figure order
The column in figure 2f & g needs to be more distinguished, as in the case in figure 5b, especially the last group (if it is white in color becomes clearer)
Line 196, I have no idea why there are such big differences between the two cultures (MGKRAS004 &5), so high like this (100 and 3.6 fold), and the authors do not explain the reason for these differences
Line 217 Klfu is not mentioned in Figure 2f, and is not mentioned in line 200 (I think it is a mistake)
Line 218 In Figure 3b, it looks like 9 miRNAs columns were downregulated, not eight, as the authors mentioned
Line 239, there is no red in Figure 3b, and it looks like magenta color
Line 308, missed the factor Myc as mentioned in the figure
Line 694 and 699, it is a double negative, not a triple negative

Author Response
Reviewer 2
We sincerely thank the reviewer for his valuable and insightful comments. Below, we provide detailed responses to each point raised, along with the corresponding corrections made in the manuscript and figures. In addition, some comments from Reviewer 2 that are not included in the list of minor comments correspond to questions that he added directly in the annotated PDF of the manuscript. For the reviewer’s convenience, we have also included these comments below the main list. They are likewise available in the attached PDF file, so he may consult them in whichever format he prefers. Finally, all modifications have been incorporated into the revised version, with changes highlighted in red within the text. A detailed, point-by-point response to each comment is provided.
Open Review
( ) I would not like to sign my review report
(x) I would like to sign my review report
Quality of English Language
( ) The English could be improved to more clearly express the research.
(x) The English is fine and does not require any improvement.
|
Yes |
Can be improved |
Must be improved |
Not applicable |
|
|
Does the introduction provide sufficient background and include all relevant references? |
(x) |
( ) |
( ) |
( ) |
|
Is the research design appropriate? |
(x) |
( ) |
( ) |
( ) |
|
Are the methods adequately described? |
(x) |
( ) |
( ) |
( ) |
|
Are the results clearly presented? |
(x) |
( ) |
( ) |
( ) |
|
Are the conclusions supported by the results? |
(x) |
( ) |
( ) |
( ) |
|
Are all figures and tables clear and well-presented? |
(x) |
( ) |
( ) |
( ) |
Comments and Suggestions for Authors
I did some corrections on the original manuscript as in the uploaded file
Minor comments
Q1. Line 86 Space [6,7]
A1. We have removed the extra space, which can now be seen in the revised line 83.
Q2a. Line 89 (id) is an incorrect abbreviation,
A2a. Thank you for your comment. Effectively, the correct way to write it is Id proteins.
This error was corrected, and the new version is:
“BMP4 supports mESC maintenance by stimulating TGF-β signaling, inducing DNA-binding/differentiation (Id) proteins, and activating SMAD1/5/8”. Line 86.
And
“RT-qPCR analysis supported this hypothesis, revealing elevated expression of key pathway-associated mRNAs—such as FZD4, CDKN1, and CCND1 (WNT); SMAD2 (Activin); SMAD5 (BMP); SMAD4 (Activin/BMP); KRAS (MAPK); and AKT2 (PI3K/AKT). Line 251-252.
Q2b. and Smad 1/5/8 is correct, where in all text mentioned smad 2/4/5
A2.b We appreciate the reviewer’s observation. The Smad notation has been corrected throughout the manuscript. The previous mention of “Smad 2/4/5” was replaced with the correct form “Smad 1/5/8” in all relevant sections of the text and figure legends to ensure consistency and accuracy. Line 86 and line 251 and 252.
Q3 Line 124 space as in the edited text
A.3. The extra parenthesis that appeared inside bracket reference 14 was removed. The sentence now is:
“PCSC+ cells overexpressed key pluripotency transcription factors—Sox2, Oct4, Nanog, Myc, and Klf4—which is consistent with their stemness profile (Figure 1d) [14]”. Line 117
Q4 Line 126 makes the tumorigenecity assay first, to be with the figure order
A4. The tumorigenic capacity of these cells was not determined; this was an unintentional error on my part, and therefore the term was removed. The new sentence is:
“Overall, flow cytometry, RT-qPCR, and Western blot, assays confirm that the triple-positive cells are PCSC+ cells.”. see line 122-123
Q5. The column in figure 2f & g needs to be more distinguished, as in the case in figure 5b, especially the last group (if it is white in color becomes clearer).
A5. Thank you for your valuable comments. The color of the bars in Figures 2f–g and 5b has been changed to improve visibility. Specifically, the updated colors can be seen in the lines 179-180 for Figures 2f and 2g, and in the lines323-324 for Figure 5b.
Q6. Line 196, I have no idea why there are such big differences between the two cultures (MGKRAS004 &5), so high like this (100 and 3.6 fold), and the authors do not explain the reason for these differences.
A6. We appreciate this valuable observation. According to the publication by Dr. Vargas’ group (Life Science Alliance, October 2023, 6(12): e202302019; DOI: 10.26508/lsa.202302019), from which these two primary cultures were obtained, the KRASMG004 primary culture harbors a mutation in the K-Ras4B gene at position G12V, whereas the KRASMG005 primary culture carries a mutation in the same gene at position G12C. Based on the literature, the G12V mutation is among the most frequent in pancreatic cancer and is associated with a poorer prognosis, higher risk of progression, and reduced survival. This mutation is characterized by aberrant activation of the MAPK signaling pathway, which promotes cell proliferation, survival, and evasion of apoptosis (1, 2).
Consequently, this mutation confers a more aggressive tumor phenotype with greater self-renewal potential, suggesting that it may support cancer stem cell maintenance. Therefore, the higher number of stem cells observed in the KRASMG004 primary culture, compared with KRASMG005, could be attributed to the presence of the G12V mutation.
References:
1.Norton C, Shaw MS, Rubnitz Z, Smith J, Soares HP, Nevala-Plagemann CD, Garrido-Laguna I, Florou V. KRAS Mutation Status and Treatment Outcomes in Patients With Metastatic Pancreatic Adenocarcinoma. JAMA Netw Open. 2025 Jan 2;8(1):e2453588. doi:10.1001/jamanetworkopen.2024.53588. PMID: 39777438; PMCID: PMC11707629.
- Budagyan K, Cannon AC, Chatoff A, Duncan JS, Uribe-Alvarez C, Chernoff J. KRAS G12V mutation-selective requirement for ACSS2 in colorectal adenoma formation. Cell Reports. 2025; doi:10.1016/j.celrep.2025.115444.
We greatly appreciate the reviewer’s insightful observation, which helped us to clarify this point in the revised version of the manuscript.
Q7. Line 217 Klf4 is not mentioned in Figure 2f, and is not mentioned in line 200 (I think it is a mistake).
A7. I am grateful for your insightful observation. Indeed, it was an error, as RT-qPCR for Klf4 was not performed at this stage of the project. Additionally, it was my mistake to include Klf4 and Myc in the description of Figure 2f; these have now been removed. The updated sentences are as follows:
Manuscript
“RT-qPCR analysis confirmed that triple-positive cells from primary cultures expressed higher levels of key pluripotency transcription factors (Oct4, Nanog, and Sox2) compared with PCSC– cells (Figure 2f), supporting their identity as PCSCs”. Lines 167-170
Figure 2f Legend:
“RT-qPCR analysis reveals higher expression of pluripotency transcription factors (Sox2, Oct4, and Nanog) in PCSC⁺ vs. PCSC⁻ primary cultures”. Lines 190-191
Q8. Line 218 In Figure 3b, it looks like 9 miRNAs columns were downregulated, not eight, as the authors mentioned
A8. We acknowledge the oversight. Indeed, nine microRNAs were found to be dysregulated. This error has been corrected, and the statement has been updated accordingly. The new statement is
“Nine of these miRNAs were consistently expressed at lower levels in PCSC+ cells compared to PCSC– cells in both the MGKRAS004 and the MGKRAS005 primary cultures (Figure 2g)”.Lines 172-174.
Q9. Line 239, there is no red in Figure 3b, and it looks like magenta color
A.9 Thank you for your valuable comment. Figure 3b was redrawn, and the color scheme was modified to improve clarity and highlight the main findings. The updated statement in the legend of Figure 3b now is as follows:
“Ten of the 31 DEmiRNAs modulate the pluripotency of pancreatic cancer stem cells through key components of the WNT (black), Activin/Nodal (magenta), BMP (green), MAPK (pink), and PI3K/AKT (blue) pathways by targeting ligands, receptors, and downstream effectors. Genes validated by RT-qPCR and/or Western blot are indicated in italic letters. Let-7b-5p and miR-24-3p are indicated as (1) and (6), respectively”. Lines 214-218.
Q10. Line 308, missed the factor Myc as mentioned in the figure
A10. We thank the reviewer again for the valuable comments and sincerely apologize for this inadvertent error. Myc has now been correctly included in the text, and the updated statement is as follows:
“Both miRNA mimics were successfully overexpressed (Figure 4a), and their individual overexpression led to reduced mRNA levels of key pluripotency factors (Oct4, Nanog, Sox2, Myc and Klf4) (Figure 4b). Lines 270-272.
Q11. Line 694 and 699, it is a double negative, not a triple negative.
A11. Indeed, it corresponds to a double-negative, not a triple-negative. This error has been corrected, and the updated version is as follows:
“miRNA from triple-positive and double-negative samples, were reverse-transcribed according to the protocol outlined by Cirera and modified by Alvarez-Hilario [52].”. Lines 659-660 and
Relative miRNA expression was calculated using the 2−∆∆Ct method [50] with normalization to RNU6 as housekeeping miRNA for triple-positive and double-negative samples, respectively.”. Lines 662-664.
Other comments from Reviewer that are not included in the list of minor comments correspond to questions he added directly in the PDF file of the manuscript. For his convenience, we have included them below the first list; however, they are also available in the attached PDF file, so he can consult them wherever he prefers.
Q1. Line 196. I have no idea why the big difference between two different cultures like this 100 and 3.6 folds
According to the publication by Dr. Vargas’ team (Life Science Alliance, October 2023, 6(12): e202302019; DOI: 10.26508/lsa.202302019), from which these two primary cultures were obtained, the primary culture KRASMG004 harbors a mutation in the K-Ras4B gene at position G12V, while the primary culture KRASMG005 carries a mutation in the same gene at position G12C.
Based on the literature, the G12V mutation is one of the most frequent in pancreatic cancer and is associated with poorer prognosis, higher risk of progression, and reduced survival. This mutation is characterized by aberrant activation of the MAPK signaling pathway, which promotes cell proliferation, survival, and evasion of apoptosis (1, 2).
Consequently, this mutation confers a more aggressive tumor environment with higher self-renewal capacity, suggesting that it may contribute to supporting cancer stem cells. Therefore, the higher number of stem cells observed in the primary culture KRASMG004, compared to KRASMG005, could be attributed to the presence of the G12V mutation.
References:
1.Norton C, Shaw MS, Rubnitz Z, Smith J, Soares HP, Nevala-Plagemann CD, Garrido-Laguna I, Florou V. KRAS Mutation Status and Treatment Outcomes in Patients With Metastatic Pancreatic Adenocarcinoma. JAMA Netw Open. 2025 Jan 2;8(1):e2453588. doi:10.1001/jamanetworkopen.2024.53588. PMID: 39777438; PMCID: PMC11707629.
- Budagyan K, Cannon AC, Chatoff A, Duncan JS, Uribe-Alvarez C, Chernoff J. KRAS G12V mutation-selective requirement for ACSS2 in colorectal adenoma formation. Cell Reports. 2025; doi:10.1016/j.celrep.2025.115444.
Q2. Lines 213 and 218. Nine not eight
We greatly appreciate the reviewer’s insightful observation, which helped us to clarify this point in the We acknowledge the oversight. Indeed, nine microRNAs were found to be dysregulated. This error has been corrected, and the statement has been updated accordingly. The new statement is
“Nine of these miRNAs were consistently expressed at lower levels in PCSC+ cells compared to PCSC– cells in both the MGKRAS004 and the MGKRAS005 primary cultures (Figure 2g)”. Line 172-174
And its legend is
“(g) Nine of the 31 DEmiRNAs are significantly downregulated in PCSC⁺ cells compared to PCSC⁻ cells in both primary cultures”. Line 191-192
Q3. check figure 3a which is compatible with 3b but different from line 232 where HIF-1 and mTOR must replace with TGF-B and BMPs to be as figure and designed figure 3a We sincerely appreciate this valuable comment.
We have revised the description of this figure to make its interpretation clearer. In the left panel of Figure 3a, all the signaling pathways that the 31 downregulated DEmiRNAs can regulate through their target mRNAs are shown, including WNT, MAPK, TGF-β, and mTOR pathways and also the pluripotency pathway of stem cells. However, only 10 out of the 31 DEmiRNAs were found to participate specifically in this pluripotency pathway. These 10 miRNAs are involved in regulating the MAPK, TGF-β, WNT, BMP, and PI3K-AKT signaling routes (Figure 3a, right panel). The pathways regulated by these 10 DEmiRNAs are the ones that correlate with those shown in Figure 3b.
To clarify this information, we added explanatory headings in Figure 3a: one on the left side labeled “31 DEmiRNAs” and another on the right side labeled “10 DEmiRNAs”. Lines 209- 210
Thus, the sentences in the text with these modifications now read as follows:
“Pathway enrichment analysis using the KEGG database revealed that the 31 DEmiRNAs are involved in multiple pathways, including cancer-related signaling, cell cycle regulation, and key transduction cascades such as WNT, MAPK, TGF-β, and mTOR (Figure 3a, left panel). In addition, these miRNAs were associated with signaling pathways regulating stem cell pluripotency (Figure 3a, left side) [17].
Focusing on this pathway, we identified ten DEmiRNAs—let-7b-5p, let-7a-5p, miR-23a-3p, miR-191-5p, let-7c-5p, miR-24-3p, let-7e-5p, miR-23b-3p, miR-103a-3p, and miR-1246—as key regulators within the pluripotency network (Table 2).
KEGG-based analysis of the predicted target mRNAs of these ten DEmiRNAs revealed multiple candidates encoding proteins involved in major signaling pathways, including MAPK, TGF-β, WNT, BMP, and PI3K-AKT (Figure 3a, right panel). These miRNAs appear to regulate a broad range of mRNAs encoding ligands, receptors, and downstream effectors within these pathways—particularly WNT, TGF-β, Nodal/Activin, BMP, MAPK, and PI3K-AKT—highlighting their potential role in modulating the signaling networks that govern stem cell pluripotency (Figure 3b, Table 3)” Lines 203-238
In addition, the legend of Figure 3a was revised to improve its clarity.
(a) mRNA targets of the 31 DEmiRNAs in PCSC⁺ cells are involved in cancer-related pathways, including WNT, MAPK, TGF-β, and mTOR, and stem cell pluripotency (left and panel. Some of these miRNAs are involved in the regulation of stem cell pluripotency through key signaling pathways, including MAPK, TGF-β, WNT, BMP, and PI3K-AKT (right panel).
Q5. It is better to concern about first 10 selected miRNA where already exist in the data base so why kept 5 miRNA and add another 5 different miRNA althoght all selected 10 already exist in data base
We sincerely appreciate this very relevant comment.
Among the 31 DEmiRNAs found to be downregulated in PCSC+ cells by microarray analysis, we randomly selected 10 miRNAs (let-7b-5p, let-7a-5p, miR-24-3p, let-7e-5p, miR-103a-3p, miR-320a, miR-107, miR-15b-5p, and miR-210-3p) for validation by RT-qPCR. The corresponding results are shown in Figure 2c for PCSC+ and PCSC− cells, as well as in primary cultures obtained from patient samples (Figure 2g). At this point, the selection of these miRNAs was performed completely at random.
Subsequently, when analyzing the possible signaling pathways in which these 31 DEmiRNAs could be involved through their predicted target mRNAs (in silico identified), we found that 10 of the 31 DEmiRNAs regulated the pluripotency pathway. Of these 10 miRNAs related to this pathway, six had already been previously included in the RT-qPCR assays shown in Figures 2c and 2g, by mere coincidence. This explains why Figures 2c, 2g, and 3c share these six miRNAs in their RT-qPCR analyses, while the other four miRNAs (miR-23a-3p, miR-191-5p, miR-23b-3p, and miR-1246) are different.
Taken together, the six common miRNAs (let-7b-5p, let-7a-5p, let-7c-5p, miR-24-3p, let-7e-5p, and miR-103a-3p), along with the four newly identified ones, are exclusively involved in the pluripotency pathway. This coincidence explains the presence of these six miRNAs in the graphs of Figures 2c and 2g, both in PCSC+ and PCSC− cells and in primary cultures derived from patient samples.
All of this information is clearly described in the revised version of the manuscript, as indicated in the corresponding lines of the text.
“To validate the microarray data, 10 of the 31 DEmiRNAs (let-7b-5p, let-7a-5p, miR-24-3p, let-7e-5p, miR-103a-3p, miR-320a, miR-107, miR-15b-5p, and miR-210-3p) were randomly selected for RT-qPCR analysis. The results confirmed the microarray findings, which revealed consistently lower expression levels of these miRNAs in the PCSC+ cells compared to the PCSC– cells (Figure 2c).” Lines 155-160
“The expression of 10 of the 31 previously validated DEmiRNAs was subsequently analyzed in both primary cultures. Nine of these miRNAs were consistently expressed at lower levels in PCSC+ cells compared to PCSC– cells in both the MGKRAS004 and the MGKRAS005 primary cultures (Figure 2g).” Lines 171-174
“Pathway enrichment analysis using the KEGG database revealed that the 31 DEmiRNAs are involved in multiple pathways, including cancer-related signaling, cell cycle regulation, and key transduction cascades such as WNT, MAPK, TGF-β, and mTOR (Figure 3a, left panel). In addition, these miRNAs were associated with signaling pathways regulating stem cell pluripotency (Figure 3a, left side) [17].
Focusing on this pathway, we identified ten DEmiRNAs—let-7b-5p, let-7a-5p, miR-23a-3p, miR-191-5p, let-7c-5p, miR-24-3p, let-7e-5p, miR-23b-3p, miR-103a-3p, and miR-1246—as key regulators within the pluripotency network (Table 2).
KEGG-based analysis of the predicted target mRNAs of these ten DEmiRNAs revealed multiple candidates encoding proteins involved in major signaling pathways, including MAPK, TGF-β, WNT, BMP, and PI3K-AKT (Figure 3a, right panel). These miRNAs appear to regulate a broad range of mRNAs encoding ligands, receptors, and downstream effectors within these pathways—particularly WNT, TGF-β, Nodal/Activin, BMP, MAPK, and PI3K-AKT—highlighting their potential role in modulating the signaling networks that govern stem cell pluripotency (Figure 3b, Table 3)”. Lines 203-238
We thank the reviewer again for this valuable comment and hope that our clarification satisfactorily addresses their concern.
miRNA mentioned only in b not a so need to correct to b
I hope that, with the previous explanation, it is now clear that this image corresponds to the right panel of Figure 3a.
Line 239 not red in figure
Thank you for your valuable comment. Figure 3b was redrawn, and the color scheme was modified to improve clarity and highlight the main findings. The updated statement in the legend of Figure 3b now is as follows:
“Ten of the 31 DEmiRNAs modulate the pluripotency of pancreatic cancer stem cells through “(b) Ten of the 31 DEmiRNAs modulate the pluripotency of pancreatic cancer stem cells through key components of the WNT (black), Activin/Nodal (magenta), BMP (green), MAPK (pink), and PI3K/AKT (blue) pathways by targeting ligands, receptors, and down-stream effectors. Genes validated by RT-qPCR and/or Western blot are indicated in italic letters. Let-7b-5p and miR-24-3p are indicated as (1) and (6), respectively”. Lines 214-218
Line 268 better to remove where it is mentioned just below
I hope that the previous clarification helps to understand why this information is included in this part of the text, as these pathways correspond to the target genes of the 10 DEmiRNAs involved in pluripotency. For this reason, this text was not removed.
Lines 277. which ten miRNA where now two sets of 10 present in fig 2 and 3 and they are different in 5 miRNA of them
I hope that the previous clarification helps to understand why this information is included in this part of the text, as these pathways correspond to the target genes of the 10 DEmiRNAs involved in pluripotency. For this reason, this text was not removed.
Line 290. where other two factors Myc and Klf4 where have same behavior as other three
Yes, as shown in the last five columns of Figure 3d, the five master transcription factors — including Myc and Klf4 — exhibit lower transcriptional expression levels in PCSC− cells than in PCSC+ cells as was expected. We have now included these five factors in the sentences. Therefore, the revised version is as follows:
“We also observed that the master pluripotency factors Oct4, Sox2, Nanog Klf4 and Myc are coregulated with target mRNAs via these signaling pathways, contributing to the maintenance of PCSCs and inhibition of differentiation (Figure 3d)”. Lines 253-256.

Round 2
Reviewer 1 Report
Comments and Suggestions for Authors
The current version is much improved. The authors have addressed mainly the comments/questions raised with new data and discussion, thereby strengthening the study and making it better suited for publication.
Comments on the Quality of English Languageok